# Ru(II) photocages enable precise control over enzyme activity with red light

Dmytro Havrylyuk [1], Austin C. Hachey[1], Alexander Fenton[1], David K. Heidary [1✉] & Edith C. Glazer [1✉]

The cytochrome P450 family of enzymes (CYPs) are important targets for medicinal chemistry. Recently, CYP1B1 has emerged as a key player in chemotherapy resistance in the treatment of cancer. This enzyme is overexpressed in a variety of tumors, and is correlated with poor treatment outcomes; thus, it is desirable to develop CYP1B1 inhibitors to restore chemotherapy efficacy. However, possible off-target effects, such as inhibition of liver CYPs responsible for first pass metabolism, make selective inhibition a high priority to avoid possible drug-drug interactions and toxicity. Here we describe the creation of light-triggered CYP1B1 inhibitors as "prodrugs", and achieve >6000-fold improvement in potency upon activation with low energy (660 nm) light. These systems provide a selectivity index of 4,000–100,000 over other off-target CYPs. One key to the design was the development of coordinating CYP1B1 inhibitors, which suppress enzyme activity at pM concentrations in live cells. The metal binding group enforces inhibitor orientation in the active site by anchoring to the iron. The second essential component was the biologically compatible Ru(II) scaffold that cages the inhibitors before photochemical release. These Ru(II) photocages are anticipated to provide similar selectivity and control for any coordinating CYP inhibitors.

[1] Department of Chemistry, University of Kentucky, Lexington, KY 40506, USA. ✉email: david.heidary@gmail.com; ec.glazer@uky.edu

Cytochrome P450s (CYPs) are a superfamily of enzymes involved in a variety of processes essential for health, including metabolism of xenobiotics and biosynthesis of important signaling molecules such as hormones[1]. However, imbalances in their activity, and/or cellular responses to their metabolic products, are important drivers in a number of disease states. A paradigm for the disease-associated activity of CYPs is CYP19A1, also known as aromatase, which generates estrogen, and thus provides the fuel for estrogen-driven cancers. When the aromatase inhibitors anastrozole, letrozole, and exemestane were introduced into clinical use, it resulted in profound extension in survival for estrogen receptor positive (ER+) breast cancer patients[2].

Several moderately selective CYP inhibitors are accepted drugs, but a major medical concern is always the possibility of off-target effects[3]. For example, systemic use of azole antifungal agents that target fungal CYP51 can result in disruption of essential steroid metabolizing CYPs and others involved in drug metabolism[4,5]. Humans express 57 active CYPs, subclassified into 18 families, and all share the same P450 fold and active site. In addition to the inherent structural and functional diversity displayed by the wild type enzymes, polymorphisms are common, with more than 1000 single-nucleotide variants known for CYP2D6 alone[6]. Thus, the vast variation of CYPs makes it challenging to predict the impact of seemingly selective CYP inhibitors in individual patients.

One option to increase control and selectivity is to develop CYP inhibitors as photolabile prodrugs, allowing for release in targeted tissues using radiation. This approach requires that the protected form be unable to inhibit the enzyme, which has proven to be surprisingly difficult. Smaller organic protecting groups are inadequate, as the linkage that provides for the photoreactivity is also subject to cleavage by CYPs. For example, coumarins, which are excellent photocages, are substrates of CYPs[7]; thus, no organic photocages have been reported for CYP inhibitors. Efforts from both our[8] and other groups[9,10] have focused on very large metal complexes as protecting groups. However, even "protected" CYP inhibitors have demonstrated significant activity. This is particularly surprising when the moiety thought to bind to the heme in the deeply buried active site is coordinated to the metal carrier, thus preventing the inhibitor's preferred binding orientation[9–12]. Moreover, the addition of the metal center and its associated ligands takes up 400–515 Å³ of space within the CYP active site or access channel. Some CYPs are able to accommodate such large groups, as was demonstrated with the various crystal structures of P450s containing bulky substrate- or inhibitor-bound moieties[13–16], but this binding requires extraordinary flexibility of the enzyme, given the buried and hydrophobic nature of the active site. As a result of these various features, photocontrol of CYP inhibitors has been an unrealized goal.

We have an interest in the cancer-associated CYP named CYP1B1. This extrahepatic enzyme is overexpressed in a variety of tumors and converts estrogen to DNA mutagens[17], so it is considered a target for chemoprevention in hormone-associated cancers. Moreover, CYP1B1 overexpression and point mutations result in resistance to many chemotherapeutics, such as cisplatin[18], daunorubicin[19], and taxanes[20–23]. Given the lack of connection between the mechanisms of action of these essential drugs, which form the foundation of the vast majority of chemotherapy regimens, CYP1B1-mediated resistance may be negatively impacting treatment outcomes for most cancer patients who receive chemotherapy. Here, we report the successful development of small molecule inhibitors of CYP1B1 based on a coordination-mediated prodrug strategy.

for various other CYPs; these so-called Type II coordinating inhibitors are used in the clinic for CYP19A1 (anastrozole and letrozole), 17A1 (abiraterone acetate), CYP11B1 (metyrapone), and fungal CYP51 (including ketoconazole, fluconazole, itraconazole, and many others). However, CYP1B1 inhibitors with coordinating groups that may be used in photocaging have not yet been developed. Moreover, addition of coordinating groups to any CYP inhibitor is a risky strategy, as off-target CYP inhibition can occur simply due to the presence of the iron-binding entity. Accordingly, we chose to base our design on a relatively simple and small CYP1B1 inhibitor, tetramethoxystilbene (TMS; 1, Fig. 1), as we could generate multiple derivatives, if needed, to establish selectivity. We predicted that only one of the methoxy-containing rings was required to engage active-site residues, which allowed for incorporation of Lewis bases at the opposite end of the molecule to form a dative bond to the iron. The coordinating group ensured proper orientation within the active site by anchoring the inhibitor in place, and the ring system distal to the heme could be the site of potential systematic modification.

For simplicity and maximal generalizability to other drug molecules, nitrogen heterocycles where chosen as the Lewis donors in 2 and 3. It was expected that the diazine ring in compound 2 would provide for superior photochemistry in the Ru(II) complex, as the less basic heterocycle was shown to create complexes with higher quantum yields for photosubstitution, $\Phi_{PS}$[24]. However, the pyridyl ring in 3 was anticipated to produce a more potent CYP inhibitor, due to its greater basicity, which

**Fig. 1 Ru(II) scaffolds and CYP1B1-selective inhibitors used in this study.** Inhibitors 2 and 3 were combined with either Scaffold I or II to give complexes 4–6. Inh = inhibitor.

## Results

**Synthesis of coordinating CYP1B1 inhibitors.** Inhibitors containing a coordinating moiety to bind the iron heme are available

increases the coordinative bond strength to the iron. Both compounds **2** and **3** were synthesized in one step using a modified Knoevenagel reaction.

**Validation of CYP1B1 coordinating inhibitors**. To rigorously assess the activity of inhibitors under biologically relevant conditions, an assay was developed for screening of CYP1B1 activity in live cells. The use of an in-cell assay, rather than the common CYP assays that utilize microsomes, was motivated by multiple factors. First, CYPs are known to engage various hydrophobic entities, such as detergents, and investigations with purified proteins, liposomes, or microsomes are likely to generate false positives, with inhibition due to non-selective interactions that would not occur in the cellular setting.

In addition, cellular uptake is a key feature for biological utility of new molecular agents, and cellular assays allow for elimination of compounds that are not cell penetrant. Finally, cell-based assays provide information on non-specific interactions that could be damaging to cell health. Given the need for CYP inhibitors to be selective and non-toxic, particularly if used as chemoprevention agents, cell-based screens provide immediate feedback on these essential characteristics.

A stable HEK cell line was created with CYP1B1 expression under the control of tetracycline to facilitate regulated and titratable expression (see the Methods section for more details). A cell line was also created for CYP1A1 as a counter screen to evaluate selectivity. CYP1A1 is the closest family member, and shares 38% sequence identity with CYP1B1[25]. The fluorogenic substrate, 7-ethoxyresorufin (REE), is a validated substrate for both CYPs and was used in both assays. Cytochrome P450 oxidoreductase (POR) is the required reductase partner for CYPs, and was also overexpressed in these systems to improve catalytic efficiency and the resulting signal for the assay.

Gratifyingly, both **2** and **3** were potent and selective inhibitors of CYP1B1 (Table 1 and Fig. 2b, c). While TMS exhibited an IC$_{50}$ (the concentration for 50% inhibition) of 83 nM, **2** was similarly potent, at 76 nM. Compound **3** was far superior, with an IC$_{50}$ of 310 pM, making it one of the most potent CYP1B1 inhibitors reported. It was also remarkably selective, with a selectivity index (SI) value of >14,000 for CYP1B1 vs. 1A1. Given the SI of 17 for TMS, this demonstrates that coordinating inhibitors can provide a ~1000-fold improvement in selectivity.

As CYP19A1 metabolizes testosterone to estradiol, and estradiol is a known substrate of CYP1B1, we reasoned that the two enzymes might share features in their active sites that could result in overlapping inhibition profiles by the new inhibitors, despite having different substrates. Thus, the activity against CYP19A1 was evaluated using a recently developed in-cell fluorometric assay[26]. While there was a small loss in selectivity for compound **2** compared to **1**, the pyridyl substituent in **3** improved the SI values over **1** by >100-fold.

The cellular assay results showed that the inhibitors were cell penetrant. The cells were >70% viable at 72 h with 10 μM compound, as quantified by metabolically active cells determined by the conversion of resazurin to resorufin (Supplementary Figs. 18–20). There were no observed modifications in cellular morphology, and it was concluded that the compounds did not induce damaging off-target effects.

Pooled human liver microsomes were used to test the potential impact of the inhibitors on multiple polymorphisms of highly promiscuous, drug metabolizing CYPs[27]. There are more than 10 different CYPs in this preparation, and the use of pooled microsomes ensures that genetic variability in CYPs is represented. While **2** was less selective than TMS, likely due to the presence of a coordinating pyrimidine, inhibitor **3** was >1000-fold

**Table 1 Inhibitory potencies and selectivity indices (SI) of stilbene derivatives.**

| Compounds | IC$_{50}$ CYP1B1 (μM)[b] | | PI | IC$_{50}$ CYP1A1 (μM)[b] | | IC$_{50}$ CYP19A1 (μM)[c] | | IC$_{50}$ phLM (μM)[d] | | SI (1B1:1A1) | SI (1B1:19A1) | SI (1B1:phLM) |
|---|---|---|---|---|---|---|---|---|---|---|---|---|
| | Dark | Light[a] | | Dark | Light[a] | Dark | Light[a] | Dark | Light[a] | | | |
| 1 | 0.083 | nd | - | 1.43 | nd | >30 | nd | 10.98 | nd | 17.2 | >360 | 132 |
| 2 | 0.076 | nd | - | 10.0 | nd | 10.71 | nd | 1.18 | nd | 175 | 188 | 16 |
| 3 | 0.00031 | nd | - | 4.57 | nd | 14.42 | nd | 0.334 | nd | 14,700 | 46,500 | 1077 |
| 4 | 0.19 | 0.012 | 15.8 | ~30 | 2.6 | 13.27 | 12.9 | 22.48 | 0.92 | ~217 | 1075 | 77 |
| 5 | 1.73 | 0.017 | 102 | 40.9 | 7.82 | 28.3 | 11.2 | >30 | ~30 | 460 | 660 | ~1750 |
| 6 | 1.9 | 0.0003 | 6333 | >30 | >30 | >30 | 18.84 | ~30 | 1.28 | >100,000 | 62,800 | 4300 |

[a]With 660 nm light, 58.7 J/cm².
[b]Determined by conversion of 7-ethoxyresorufin (REE).
[c]Determined by conversion of dibenzylfluorescein (DBF).
[d]Determined by conversion of 7-benzoyloxy-4-trifluoromethylcoumarin. The selectivity indices were calculated as the ratio of the IC$_{50}$ values for different CYPs to CYP1B1 (for complexes **4–6** the IC$_{50}$ (light) values were used). (n = 3).

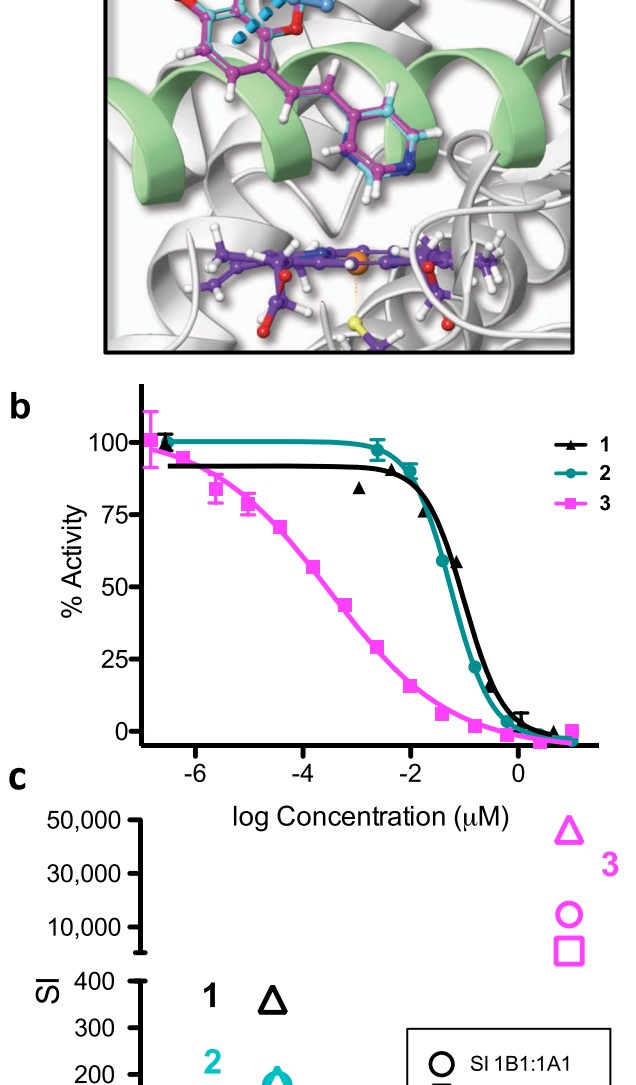

**Fig. 2 Heterocycle containing inhibitors for CYP1B1. a** Docking of **2** and **3** shows the position of the heterocycle over the heme (purple) and key contacts with Phe231. Docking based on PDB 3PM0. **b** Dose–responses in CYP1B1 activity assay ($n = 3$ independent samples; the error bars correspond to the standard deviation of the three replicates). **c** Comparison for **1–3** of selectivity index (SI; for three CYP systems indicated in the inset) and potency in CYP1B1. The $IC_{50}$ (in μM) is shown as the negative log to enable comparison over a wide range.

selective for CYP1B1 (Table 1 and Fig. 2c). This is notable, as the liver CYPs have evolved to bind the widest possible variety of molecules.

To provide context, we compared these results to those reported for anastrozole, a drug used by breast cancer patients for durations of 5–10 years as a maintenance therapy. Anastrozole is

at least 500-fold selective for CYP19A1 over liver CYPs[28]; this reflects an upper limit for extremely selective CYP inhibitors. In contrast, CYP inhibitors such as ketoconazole are used with great caution, as their selectivity is poor. Ketoconazole's selectivity for its target, fungal CYP51, is only ~2-fold compared to the human isoform of CYP51 ($K_d$ values of 27 and 42 nM)[29], and it is a potent inhibitor of CYP3A4 ($IC_{50} = 40$ nM), as well as members of the 1B, 2B, and 2C families[30]. Thus, the novel CYP1B1 inhibitors reported here provide SI values that rival or exceed the most selective CYP inhibitors used in the clinic.

**Computational studies.** The closely related structures of TMS, **2**, and **3**, but their wide range of potencies in CYP1B1 motivated computational studies to rationalize structure-activity relationships. However, simple molecular docking experiments were unable to identify significant differences in the protein-ligand complex, as the top predicted binding poses overlayed, as shown in Fig. 2a, with a calculated RMSD of only 0.1 Å. Short molecular dynamics trajectories were completed, and while some crucial residue contacts are shared, compounds **2** and **3** diverge in their behavior over the 20 ns trajectories. Compound **3** immediately produced a relatively stable RMSD of 1.67 Å, while **2** exhibited wide fluctuations for the first half of the trajectory before converging on a semi-stable binding pose at an RMSD of 2.52 Å (Supplementary Fig. 1). The more dominant interactions in the trajectory of **3** include hydrophobic contacts with Val126 and Ala330, π-stacking with Phe134, and the formation of a water bridge with Ser127 (Supplementary Table 1 and Supplementary Fig. 46). Key interactions in the trajectory of **2** include the formation of transient water bridges with Gly329, Thr334, and a mixture of water bridge formation and hydrophobic contacts with Ile399. Both molecules engage in π-stacking interactions with Phe231 and Phe268 with comparable duration. However, differences in pKa values for methyl pyrimidine vs. methyl pyridine (2.0 vs. 5.9; from Scifinder) may provide a partial explanation for the variation in activity, as the greater basicity of **3** should result in a stronger coordinative bond, and thus, improved potency. This hypothesis is related to the prevalence of the 1,2,4-triazole ring in clinical CYP inhibitors, in contrast to the less basic 1,2,3-triazole ring[31].

**Mutational studies.** Site directed mutagenesis was used to probe the importance of specific active-site contacts identified through the structural assessment (Fig. 3a, b) and simulations. In all cases, conservative mutations were made to maintain polar or hydrophobic character and approximate size. The aromatic residues that frame the inhibitor binding site, Phe143 and Phe231, were mutated to Leu to determine the importance of the π interactions. In addition, the Ser127 residue, located in the upper corner of the active site on the B-C loop, was chosen for mutation based on the simulations. This amino acid transiently contacts the inhibitors, but also forms a hydrogen bond with Asp326 of the I-helix, and thus may impact the position of this helix with regards to the B-C loop. The Ser269 residue, located in the G helix at the top of the active site, was anticipated to form polar contacts with the inhibitor side chains, if the inhibitor disengaged from the heme and slid away against the I-helix. Finally, two other active-site amino acids, Ala330 and Thr334, are important contacts for inhibitors (Supplementary Table 1). Rather than directly mutating these residues in the I-helix, neighboring residues Gln332 and Asp333 were targeted, with the hope of interrupting amino acid interactions that position the helix within the active site.

While the Phe231Leu mutant was expressed, it was inactive. This supported the premise that Phe231 is responsible for orienting and stabilizing contacts with small molecules, as this

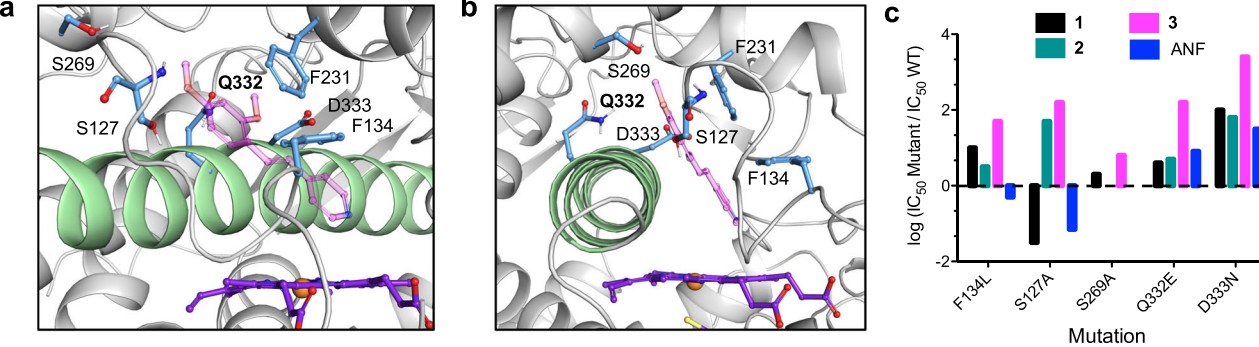

**Fig. 3 Mutations reveal key contacts in CYP1B1. a, b** Location of mutated amino acids. The I-helix is shown in green, the heme in purple, and **3** (lavender) was docked in the structure. Q332 (bold) is not found in other CYP1 family members. F231 is essential for turnover of the REE substrate. **c** Effect of point mutations on the efficacy of CYP1B1 inhibitors. The inhibitors were less potent in the mutants, with the exception of ANF and **1** in the S127A mutant and ANF in the F134L mutant. The log of the ratio of $IC_{50}$ values was used to simplify visualization. See Supplementary Table 2 for all values.

interaction would be essential for the binding of the 7-ethoxyresorufin substrate. All the other mutants were active, allowing for investigations into the impact of each amino acid on inhibitor efficacy. Both TMS (**1**) and α-naphthoflavone (ANF), a polyaromatic CYP1 family inhibitor, were used as controls. While ANF binding is dominated by hydrophobic contacts, **1** was anticipated to be able to engage in some polar interactions, but not metal coordination.

The mutants exhibited a range of $IC_{50}$ values with the different inhibitors, as shown in Supplementary Table 2. The ratio of $IC_{50}$ values for the WT vs. the mutant are shown in Fig. 3c and reflects the importance of the specific contact; the log of the value is used to aid in visualization. The Ser269Ala mutation had relatively little effect, and the Phe134Leu mutant, which eliminated a predicted edge-to-face aromatic interaction with inhibitors, had a moderate impact. This highly conserved sidechain, analogous to Phe123 in CYP1A1, is located on the interior face of the active site and is also implicated in orientation of substrates.

The mutations that had the largest impact on inhibitor potency were Ser127Ala, Gln332Glu, and Asp333Asn, with a decrease in activity of 2–3 orders of magnitude. While the Ser127 residue was not expected to create long-lived contacts with the inhibitors, we hypothesized that the removal of the H-bond with Asp326 could shift the I-helix. The Asp326 sidechain also engages in a H-bond with ANF, which may be strengthened by the replacement of the Ser127 with an alanine; this could also explain the surprising increase in potency for compound **1**. However, the mutation has the opposite effect for **2** and **3**, demonstrating a change in the molecular factors regulating their binding.

Gln332Glu and Asp333Asn are located on the key I-helix, which transverses the active site and against which the substrates and inhibitors rest. Asp333 forms a salt bridge with Lys512 of β sheet 4 to regulate tertiary structure; this interaction is conserved in the CYP1 family. In contrast, a phenylalanine is found in CYP 1A1 in the position analogous to Gln332. The importance of the contacts made between **2** and **3** and these amino acids may be the key to the extraordinary selectivity observed for these inhibitors for CYP1B1 over CYP1A1.

**Synthesis of photocaged CYP1B1 inhibitors.** We recently synthesized and assessed a variety of Ru(II) scaffolds in order to identify a suitable inorganic system that was effective for photocaging enzyme inhibitors[32]. One structure provided the desired biocompatibility, thermal stability, and ability to be triggered with visible light (from 450–660 nm; Scaffold II, Fig. 1). This Ru(II) photocage incorporated a 2,2'-biquinoline ligand, which shifted the absorption profile to longer wavelengths[32,33]. This change in

absorption is due to the lower energy of the metal to ligand charge transfer (MLCT) transitions, which depend on the lowest unoccupied molecular orbital (LUMO) of the conjugated biquinoline ligand. Owing to the extended conjugation of quinoline, the transition is bathochromically shifted from analogous 2,2'-bipyridyl systems. The use of longer wavelengths of light are appealing for the ability to achieve greater depths of penetration into tissues, but it was also an important design feature for the complexes, as stilbene systems undergo *trans* to *cis* photoisomerizations with high energy (generally UV) light[34].

Additional optimization included incorporation of carboxylic acids to the biquinoline. The [2,2'-biquinoline]-4,4'-dicarboxylic acid ligand (also known as bicinchoninic acid, bca) reduced cellular toxicity caused by the metal complex, such that no adverse effects were observed up to 100 μM concentrations[32]. To occupy three of the remaining coordination sites in the octahedral complex, the tridentate 2,2';6',2"-terpyridine (tpy) ligand was added. This left one site available for coordination of the active inhibitor, which is a monodentate ligand.

Compounds **2** and **3** were combined with Scaffolds I and II to create photocaged CYP1B1 inhibitors, resulting in octahedral Ru(II) polypyridyl coordination complexes **4**–**6** (Fig. 1). These structures are contrasted to previous photocaged enzyme inhibitors designed using the Ru(bpy)$_2$ scaffold (bpy = 2,2'-bipyridyl)[8,35–37], which allows for incorporation of one or two inhibitors for each molecular component. While combinations of tridentate ligands and strain-inducing bidentate ligands creates stoichiometric photocages, the advantage is that these systems have more predictable photochemistry[38].

**Evaluation of Ru(II) photocages.** As shown in Supplementary Fig. 2, the absorption profiles of complexes **4**–**6** varied as a function of the monodentate "caged" ligand (compound **5** vs **6**) and the bidentate ligands (compound **4** vs. **5**). Both mono- and bidentate ligands had an impact on the longest wavelength absorption peak, $\lambda_{abs}$, and extinction coefficient (ε) values (Table 2). Complexes **4**–**6** exhibit MLCT maxima between ~530 and 545 nm in H$_2$O, with tailing absorption out to 650 nm. The coordination of the 4-substituted pyridyl ligand induced a bathochromic shift of the MLCT by 15 nm, and a tail that extended to 700 nm, facilitating activation with low-energy light, as shown in Fig. 4a.

In order to standardize photochemical evaluation parameters and to compare the photosubstitution for **4**–**6** with reference compounds studied previously, the quantum yield for photosubstitution, $\Phi_{PS}$, was determined in H$_2$O using 470 nm light. As anticipated, variation in yields was observed, with $\Phi_{PS}$ ranging from 0.0004 to 0.055 (Table 2). The coordination of pyrimidine-based ligand **2** to

**Table 2 Thermal stability, photophysical, and photochemical properties of compounds 4–6.**

| Compounds | $\lambda_{abs}$, nm, ($\varepsilon$ $(M^{-1}cm^{-1})$)[a] | $\Phi_{PS}$[b] | Stability[c] |
|---|---|---|---|
| 4 | 395 (23,900); 455 (15,800); 535 (10,900) | $0.055 \pm 0.003$[d] | 43 |
| 5 | 400 (13,000); 455 (8,300); 540 (6,300) | $0.02 \pm 0.002$[d] | 50 |
| 6 | 380 (21,500); 455 (7,300); 550 (6,600) | $0.00043 \pm 0.00005$[e] | 98.6 |
|   |   | $0.0081 \pm 0.0006$[f] |   |

a In $H_2O$.
b Irradiation wavelength, $\lambda_{irr}$ = 470 nm.
c Determined as % remaining at 24 h (37 °C) in $H_2O$.
d Quantum yield for photosubstitution, $\Phi_{PS}$, in 5% DMSO in $H_2O$, calculated by optical approach ($n = 3$). DMSO was added to improve the solubility of 6, so it was also used for the other complexes.
e $\Phi_{PS}$ in 5% DMSO in $H_2O$, determined by HPLC approach. HPLC was used due to overlap in absorbance profiles for 6 and its photochemical product.
f $\Phi_{PS}$ determined in MeCN by HPLC.

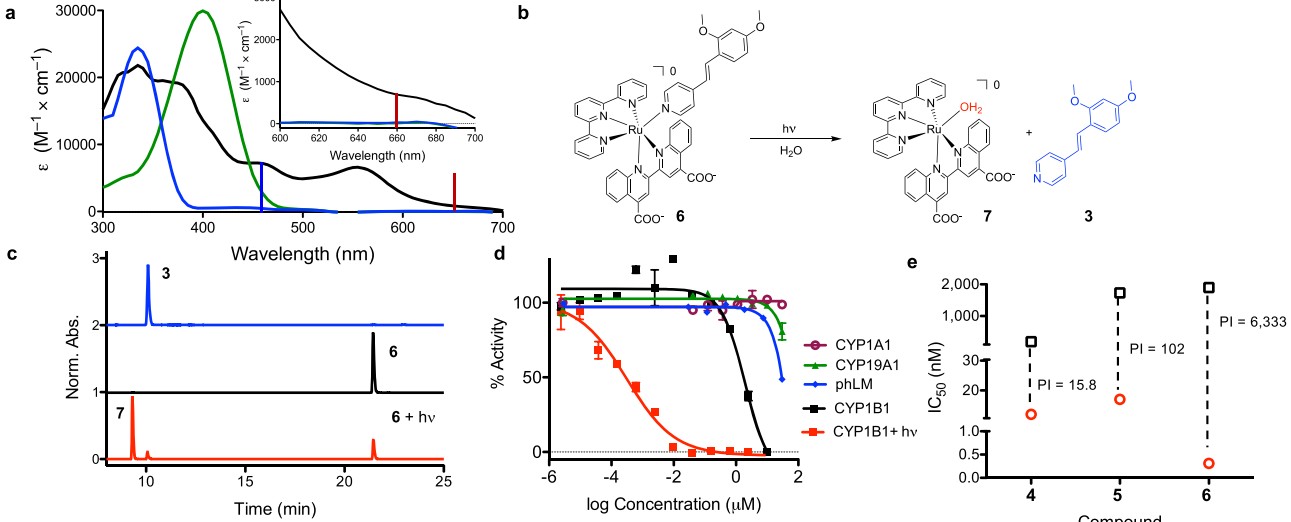

**Fig. 4 Photocontrol of a Ru(II) complex containing a CYP1B1 inhibitor. a** Absorption spectra of **6** (black, solid line) in phosphate buffered saline (PBS), **3** in MeCN (blue, solid line), and **3** in MeCN following protonation with 1% formic acid (green, solid line). The excitation wavelengths for the blue and red LEDs are indicated with vertical bars. Inset shows an expansion of the 600–700 nm region. **b** Photoejection scheme for **6** in water. **c** HPLC analysis showing the chromatograms for **3**, **6**, and **6** following irradiation with 660 nm light (58.7 J/cm²). The formation of complex **7** and inhibitor **3** were verified by UV/Vis (Supplementary Fig. 11). The protonated form of **3** was observed due to the presence of acid in the HPLC mobile phase. Detection wavelength = 280 nm. **d** Dose–responses for **6** for enzyme inhibition in the dark with CYP1A1, 19A1, and pooled human liver microsomes (phLM), CYP1B1, and with 1B1 following irradiation with 660 nm light (58.7 J/cm²) ($n = 3$; the error bars correspond to the standard deviation of the three replicates). **e** Comparison of potency and PI values (the ratio of the IC₅₀ for enzyme inhibition in the dark vs. light) for Ru(II) complexes **4**–**6** (IC₅₀ in the dark (black squares) and activated with 660 nm light (red circles)). This data demonstrates a range of 10³ due to improvement in both the inhibitor, with a shift to lower IC₅₀ values, and Ru(II) scaffold, with the increase in the IC₅₀ in the dark.

Ru(II) scaffolds (to form compounds **4** and **5**) resulted in up to 100-fold higher $\Phi_{PS}$ compared with complex **6**; the higher $\Phi_{PS}$ values are consistent with the previous studies comparing the photochemistry of Ru(II) complexes with pyridine and diazines ligands[32]. The $\Phi_{PS}$ for **6** is also environmentally sensitive, with higher values in less polar environments (Supplementary Table 7), which may be due to improved solubility. An alternative explination would be that the environment alters the energy of the ³MLCT state, and this impacts the equilibration with the ³MC state.

The stability of each complex was assessed over 24 h under aqueous conditions at 37 °C (Supplementary Figs. 12–15). The compounds with the higher $\Phi_{PS}$ (**4** and **5**) exhibited slow degradation over 24 hr, while complex **6** remained stable, with less than 2% degradation in the presence of glutathione or imidazole, and at low pH (Table 2 and Supplementary Fig. 16). A control compound, [Ru(tpy)(bca)(pyridine)], **8**, was also investigated, and was stable over 72 h (Supplementary Fig. 17, <7% degradation). Given the potency of the caged inhibitor, thermal stability is of critical importance, making complex **6** the preferred photocaged candidate.

**Photoactivated CYP inhibition**. Both compounds **4** and **5** were effective in providing photocontrol over inhibitor delivery, with photoactivity indices (PI; the ratio of the IC₅₀ values in the dark and light) of 16–102 ($\lambda_{ex}$ = 660 nm). These results were highly promising for the complexes as photocaged enzyme inhibitors. However, the potency of the complexes in the dark (0.2–2 μM) was striking. While the metal complexes may hydrolyze and dissociate the inhibitors under the cellular conditions, we do not believe this is the primary factor driving the inhibitory potency in the dark, especially for compound **6**, that possesses high purity (Supplementary Fig. 44) and stability over 72 h of incubation (Supplementary Figs. 15 and 16). Inhibition of CYP1B1 as a result of binding to the protein in a non-specific manner, which could occur with purified systems, was ruled out as the experiments were performed in a cell-based assay, where there are many other potential hydrophobic binding partners. It is plausible that the caged inhibitor interacts with some important surface region, such as the P450 oxidoreductase binding site. As all CYPs share the same fold, this could result in non-specific inhibition of other CYPs. However, compounds **4** and **5** had no impact on CYP19A1

or CYP1A1 at concentrations ≥10 μM. The control compound, **8**, which has the same scaffold as **5** and **6**, but releases pyridine, also had no impact on any CYP at concentrations up to 30 μM both in the dark and following irradiation to form **7** (Supplementary Fig. 24). Thus, the engagement with the enzyme is selective both with regards to the enzyme target and the presence of the inhibitors in the Ru(II) structure. Moreover, model complex **8** had no effect on cell health, as shown in Supplementary Fig. 24d, validating the use of the Ru(II) scaffold for photocages.

Some Ru(II) complexes can generate singlet oxygen when photoexcited, a feature that has been applied to the selective photoinactivation of proteins[39]. In this approach, the Ru(II) system is conjugated to a peptide that is a specific ligand for the protein target; binding of the peptide to the protein ensures close proximity of the Ru(II) center and thus, upon photoexcitation, the generation of a high local concentration of singlet oxygen. To test if this was occurring, the ability of compounds **4–8** to photosensitize singlet oxygen was investigated (Supplementary Fig. 27). As very little of this reactive oxygen species was generated, inactivation of the CYP enzymes by this process appears unlikely.

The best photocaged CYP1B1 inhibitor, compound **6**, was created by incorporation of **3** into scaffold II. The complex could be activated with low-energy light (Fig. 4a), cleanly ejected **3** (Fig. 4b, c) and produced complex **7**, which was biologically inert. As with **4** and **5**, the intact complex exhibited inhibition of CYP1B1, but had little effect on other CYPs up to concentrations of 30 μM (Table 1 and Supplementary Figs. 21–23). Photoremoval of the Ru(II) protecting group with 660 nm light resulted in an $IC_{50}$ of 300 pM for inhibition of CYP1B1, which corresponded to a PI value of >6300. This represents the largest PI value for any Ru(II) photocage, by 10–1000-fold, depending on the system. We are not aware of other inorganic or organic photocages that provide this level of photocontrol.

The best combination of co-ligands used in the Ru(II) photocage is vital to the creation of biocompatible systems that can be activated with low-energy light. Addition of carboxylic acids to the biquinoline ligand reduced inhibition of the CYP1B1 enzyme by the complex by ~10-fold (complex **5** vs. **4**), and concentrations ≥10 μM were needed to observe interactions with other CYPs (Table 1 and Supplementary Fig. 22). However, to achieve >$10^2$-fold differences in activity based on photocontrol required the creation of more potent inhibitors, with activity at low- to sub-nM concentrations. We believe this may be a consistent feature for light-activated P450 inhibitors until a photoactive Ru(II) scaffold is found that is biologically inert and does not allow for interactions with proteins.

**Impact on protein stability**. To probe the impact of these inhibitors on the biophysics of CYP1B1, the thermal stability of the recombinant enzyme was assessed and compared with ANF. Circular Dichroism (CD) was used to monitor changes in the secondary structure with increasing temperature. ANF is known to bind with its flat surface resting against the helix I between Gly329 and Ala330, and π-stacked with Phe231[25]. The thermal stability of CYP1B1 increased by 3 °C with 10 μM ANF; a 2 °C stabilization was observed with inhibitor **3** (Table 3 and Supplementary Fig. 26). A slight decrease in stability was seen with **6** at a concentration of 20 μM. Thus, while the intact metal complex is able to inhibit the enzyme, it is not binding in a manner that enhances stability. If the complex bound to the exterior of the protein, potentially at the POR interface, this would be likely to cause inhibition of other CYPs, which is not observed. Thus, we believe it appears most likely that the complex occupies the mouth of the active-site channel, or the active site.

**Table 3 $T_m$ values for recombinant CYP1B1+/− inhibitors.**

| Condition | Concentration | $T_m$ (°C)[a] |
|---|---|---|
| No compound | - | 47.9 +/− 0.25 |
| ANF | 10 μM | 51.0 +/− 0.3 |
| **3** | 10 μM | 50.4 +/− 0.26 |
| **6** | 20 μM | 47.0 +/− 0.4 |
| **8** + $h\nu$[b] | 20 μM | 47.7 +/− 0.51 |

[a]$T_m$ values were calculated from curve fit of a minimum of two replicates. The reported error is the standard deviation of the fit relative to the experimental values.
[b]Red light (660 nm, 58.7 J/cm²).

## Discussion

Creation of selective, stable, and responsive agents for photopharmacology is quite challenging. Photoswitching compounds, which provide the ability to reversibly turn "on" and "off" the activity of a variety of molecules, thereby enabling dynamic studies, commonly exhibit ~10-fold differences between their active and inactive forms[40,41]. Photocages, which are irreversible, tend to have higher PI values, but the best organic systems usually require cell damaging high-energy light, and PI values remain below 100. Inorganic systems provide advantages that include use of longer wavelengths of light, and Ru(II) has emerged as the most promising protecting group[36,42,43]. In a recent report, five different light-activated Ru(II) complexes containing CYP3A4 inhibitors were designed and evaluated, and the highest PI value achieved was 2.1, highlighting the difficulty of this goal. Indeed, the intact Ru(II) complexes often had greater activity than the released inhibitor[10]. Surprisingly low PI values are a persistent problem for Ru(II) photocages, with dark $IC_{50}$ values commonly being in the μM range. For example, PI values are generally below 40 for caged inhibitors of cysteine proteases[35,44,45], nicotinamide phosphoribosyltransferase (NAMPT)[46], tubulin polymerization[47], and CYPs[9].

Our studies also demonstrate that large Ru(II) complexes appear able to inhibit CYPs, and establish that this inhibition occurs with notable selectivity in cells, despite the presence of many competing biomolecules. As anticipated, given their size, complexes **4**, **5**, and **6** could not be docked into the CYP1B1 structure, so structure-based modifications of the Ru(II) center to reduce binding is not possible. What remains to be determined is what drives the interaction of these, and other[10], metal-coordinated CYP inhibitors with such extraordinary affinity and selectivity to their target enzymes. As the metal center blocks key components of the inhibitor, and the enzyme must distort into a thermodynamically unfavored conformation to accommodate this unnatural system, these metal complexes force us to reassess what we assume we know about CYP inhibitors.

In this work, we have achieved precise photocontrol for selective inhibition of CYP1B1, and established a strategy whereby CYP engagement can be modulated through molecular components of both the photocage and the inhibitor. Rational redesign of CYP1B1 inhibitors resulted in new compounds that follow Lipinski's rules (Supplementary Table 8), and an agent with pM $IC_{50}$ values for CYP1B1 and selectivity over other CYPs of 3–4 orders of magnitude. This molecule, combined with an optimized Ru(II) caging scaffold, yielded the best system, which possessed protein inhibition at pM concentrations when irradiated with red light. To the best of our knowledge, **3** is the most selective reported CYP inhibitor, and **6** provides the highest photoactivity index described for an organic or Ru(II)-based photocaged enzyme inhibitor.

## Methods

Full and more detailed experimental information can be found in the Supplementary Information.

## Synthesis

*General.* All materials were purchased from commercial sources and used without any further purification. All $^1$H-NMR and $^{13}$C-NMR were obtained on a Varian Mercury spectrometer (400, 100 MHz) and chemical shifts are reported relative to the residual solvent peaks. Electrospray ionization (ESI) mass spectra were obtained on a Thermo Scientific Q-Exactive Orbitrap mass spectrometer equipped with a heated electrospray ionization source at the University of Kentucky Mass Spectrometry Facility (UKMSF). UV/Vis absorption spectra were obtained on a BMG Labtech FLUOstar Omega microplate reader. Light activation experiments were performed using a 470 and 660 nm LED array from Elixa. For the 660 nm LED, peak emission $\lambda_P = 660$ nm, spectral line full width at half-maximum $\Delta\lambda = 20$ nm (see Supplementary Fig. 45). The Prism software package was used to analyze data.

*HPLC analysis for purity and photoejection products.* The purity of Ru(II) complex and photoejection products were analyzed using an Agilent 1100 Series HPLC equipped with a model G1311A quaternary pump, G1315B UV diode array detector and Chemstation software version B.01.03. Chromatographic conditions were optimized on a Column Technologies Inc. C18(2), 100 Å (250 × 4.6 mm inner diameter, 5 μM) fitted with a Phenomenex C18 (4 × 3 mm) guard column. Injection volumes of 20 μL of $30 - 100$ μM solutions of the complex were used. The detection wavelength was 280 nm, except when otherwise noted. Mobile phases were: mobile phase A, 0.1% formic acid in diH$_2$O; mobile phase B, 0.1% formic acid in HPLC grade acetonitrile. The mobile phase flow rate was 1.0 mL/min. The following mobile phase gradient was used: $98-5\%$ A (containing $2-5\%$ B) from 0 to 5 min; $95-70\%$ A ($5-30\%$ B) from 5 to 15 min; $70-40\%$ A ($30-60\%$ B) from 15 to 20 min; $40-5\%$ A ($60-95\%$ B) from 20 to 30 min; $5-98\%$ A ($95-2\%$ B) from 30 to 35 min; reequilibration at 98% A (2% B) from 35 to 40 min.

*Counter-ion exchange.* Prior to photoejection studies and biological testing, counterion exchange was performed on compound **4**. The PF$_6^-$ salt of **4** was converted to the Cl$^-$ salt by dissolving 10 mg of product in 2 mL methanol. The dissolved product was loaded onto an Amberlite IRA-410 chloride ion exchange column, eluted with methanol, and the solvent was removed in vacuo.

## Photoejection studies

Quantum yields for the complexes **4–6** were determined by optical and HPLC approaches, since there are advantages to both, as described previously[24,32]. For the optical approach, the Ru(II) complexes were analyzed in a 96-well plate at a final concentration of $30-50$ μM and a path length of 0.5 cm. Scans were taken at set time points for 240 min. In all cases, the light source was a 470 nm LED array from Elixa. The photon flux of the lamp for irradiation in the plate was determined by ferrioxalate actinometer (1.77E-8 Mol/s). The absorbance of complexes at 470 nm ranged from 0.12 to 0.36, with photon absorption probability (F) from 0.22 to 0.42. Therefore, the moles of photon absorbed have been calculated as the product of photons irradiated and photon absorption probability.

For compound **6**, the UV/Vis of the product in aqueous solutions was very similar to the starting material. Accordingly, the quantum yield for photosubstitution of complex **6** in MeCN and MeOH were determined by optical methods, and were also calculated in MeCN and H$_2$O based on HPLC analysis. The Ru(II) complex was irradiated in a quartz cuvette at a final concentration of 40 μM and a path length of 1 cm. The photon flux of the lamp for irradiation in cuvette was determined by ferrioxalate actinometer (2.32E-7 Mol/s). The absorbance of complex at a concentration of 40 μM at 470 nm was 0.27 with photon absorption probability (F) of 0.46.

## Enzyme activity studies

*In cell activity assays for CYP1B1, 1A1, and 19A1.* The genes for CYP1B1, CYP1A1 and CYP19A1 were purchased from Origene. To allow for maximal turnover, the pcDNA4 T/O vector was modified for dual expression with P450 oxidoreductase (POR). Both the CYP and POR expression was under the control of the TetO$_2$ inducible CMV promoter[26]. In brief, CYP1B1, CYP1A1 or CYP19A1 was cloned into pcDNA4 T/O. AgeI and BsiWI restriction sites were then incorporated into the plasmids. The promoter region from pcDNA4 T/O was amplified and ligated, creating a second TetO$_2$ inducible CMV promoter with the pcDNA4 T/O multiple cloning sites (MCS). POR was cloned into this MCS using the KpnI and XhoI restriction sites.

Cell lines were generated in the HEK293 T-Rex cell line. Following transfection, cells were selected with 500 μg/mL Zeocin, and 7.5 μg/mL Blasticidin to create a stable pool. Inducible expression of CYP and POR was confirmed by immunoblot. The cell lines were maintained in DMEM supplemented with 10% FBS, 100 U Penicillin and 100 μg/mL Streptomycin at 37 °C with 5% CO$_2$. The lines were maintained with 250 μg/mL Zeocin and 7.5 μg/mL Blasticidin to ensure continued selective pressure.

Cell lines were seeded onto Geltrex coated 96-well plates at 40,000 cells/well, and grown overnight in DMEM media containing 1 μg/mL tetracycline. The media was then replaced with Opti-MEM supplemented with 2% FBS and 1 μg/mL tetracycline. Compounds were serially diluted in Opti-MEM supplemented with 2% FBS and 1 μg/mL tetracycline and an equal volume of compound added to the cells ($n = 3$). Following a 1 h incubation, resorufin ethyl ether (REE; for CYP1B1

and 1A1) or dibenzylfluorescein (DBF; for 19A1) was added to the cells for a total concentration of 5 μM for REE and 1 μM for DBF. Time points were taken over a period of 24 h using a Spectrafluor Plus plate reader (Tecan) with an excitation wavelength of 530 nm and emission wavelength of 595 nm for CYP1B1 and CYP1A1, and excitation wavelength of 480 nm and emission wavelength of 530 nm for CYP19A1. Dose–response curves were generated in Prism and normalized using the no-compound control (set as 100% activity) and the no-cell control (set as 0% activity; this value in the raw data reflects the background fluorescence of the dye alone).

For the light-activated compounds, the experiment was modified as follows: Cells were seeded and grown overnight as described above. Media was replaced with L-15 containing 1 μg/mL tetracycline; L-15 media was used in place of opti-MEM™ to maintain CO$_2$ outside the incubator, and to prevent cellular damage from light. Compounds were serially diluted in L-15 with 1 μg/mL tetracycline and added to the cells followed by a 1 h incubation. The cells were then exposed to red light for 1 h (660 nm, 58.7 J/cm$^2$) followed by the addition of an equal volume of Opti-MEM supplemented with 4% FBS and 1 μg/mL tetracycline. REE or DBF was added as described above.

*Cellular viability.* The parental HEK T-Rex cell line was maintained as above, in the absence of Zeocin. Compounds were serially diluted in media, and the cells were dosed from 0–30 μM compound, and then incubated at 37 °C ($n = 3$). Following incubation, resazurin was added to each well (80 μM) and cell viability was quantified by the conversion of resazurin to resorufin using an excitation wavelength of 535 nm and detecting emission at 595 nm. For cell viability studies following irradiation, L-15 media was used in place of opti-MEM™. The cells were incubated with the compound in the dark for 30 min prior to light exposure, using the same parameters as in the enzyme activity assay, and then treated as above. Dose–responses were generated in Prism and normalized using the no-compound control (100% viable) and the no-cell control (0% viable).

*Analysis in pooled human liver microsomes (phLM).* Commercial preparations of pooled human liver microsomes come from 50 individuals, providing representation of biologically relevant CYP variants. Compounds were screened for inhibition of cytochrome P450 enzymes in phLMs using a 96-well kinetic assay. The final concentration of DMSO did not exceed 0.8%. Compound stocks were diluted in dose–response, and added in 100 mM potassium phosphate buffer pH 7.8 with 50 μM 7-benzoyloxy-4-trifluoromethylcoumarin (Corning, prepared as a 10 mM DMSO stock) and 16 μg of phLMs (XenoTech, prepared as 20 mg protein/mL suspended in 250 mM sucrose) per well ($n = 3$). The compounds were incubated for 5 min in the absence of light at 25 °C. The 96-well plate was then either protected from light or irradiated with red light for 1 h (660 nm, 58.7 J/cm$^2$). Enzymatic turnover was initiated upon addition of NADPH (TCI, prepared immediately prior to addition) at a concentration 1.2 mM per well. To determine background fluorescence from the assay, NADPH was omitted from several control wells. The reactions were immediately placed in a SpectraFluor Plus plate reader (Tecan) preheated to 37 °C, and fluorescence measurements taken every 3 min for 120 min with excitation of 435 nm and emission of 525 nm. Fluorescence measurements in the absence of NADPH were subtracted from those in the absence of NADPH, and the data was processed with GraphPad Prism 6. The linear portion of the signal for each dose point with respect to time was first fitted to a line to determine the rate. The resulting slope of these lines was then plotted against the logarithmic concentration of the study compound to provide a dose–response curve.

## CYP1B1 stability

CYP1B1 was expressed and purified as reported[25]. The purified enzyme exhibited the classical shift to the 450 nm form in the presence of carbon monoxide, and demonstrated spin shifts in the presence of inhibitors. The stability of CYP1B1 in the presence of several inhibitors was determined by temperature melt, with the change in ellipticity monitored at 230 nm by circular dichroism with a J-815 Spectrometer (Jasco). The enzyme was prepared at a concentration of 0.1 mg/mL in 20 mM Potassium Phosphate pH 7.4, 300 mM NaCl, 10 mM CHAPS, 20% Glycerol with 1 mM 2-mercaptoethanol. Inhibitors were incubated with the enzyme at 25 °C for 10 min. Spectra were taken scanning from 200 to 260 nm in duplicate. The temperature was increased followed by a 3 min equilibration period before spectra acquisition. The ellipticity values at 230 nm were plotted against temperature and fit to determine the $T_m$ (Prism).

## Docking and MD experiments

For the computational investigation of CYP1B1, the PDB structure 3PM0 was used, which has the inhibitor α-naphthoflavone (ANF) bound in the active site[25,48]. To prepare the published coordinates for computation, Maestro was used to process and refine the structure. The need for preparing protein crystal structures for computation is well established and includes manipulations not performed in the X-ray crystal structure refinement stage of data collection[49]. These preparation steps include, but are not limited to, assignment of bond orders, addition of hydrogen atoms to the structure, optimization of the hydrogen bonding (H-bonding) network, resolution of atomic clashes, processing residues with missing electron density, and minimization of the protein structure. Zero-order bonds to metals were created, missing side chains[50] and loops[51] were filled with Prime, and waters farther than 5 Å from heteroatom

groups were removed. After this initial processing, the heme iron atom was set to a charge state of +3 to reflect the resting ferric form of the heme.

To test the ability of Maestro to reproduce key interactions, the bound ANF inhibitor was removed from the active-site cavity, and then successfully docked using the Glide application[52–54] to an RMSD of 0.34 angstroms from its original position.

To further evaluate the active-site model for its ability to predict interactions, the CYP1 family substrate 7-ethoxyresorufin was docked into the CYP1B1. This molecule is the fluorogenic substrate used in the ethoxyresorufin-O-deethylase (EROD) assay[55], which relies on oxidative dealkylation to form resorufin as a measure of enzymatic activity. Docking results predict placement of the critical oxygen atom near the catalytic heme prosthetic group, while the π-system of the resorufin core is stabilized by nearby phenylalanine residues. These results support the fidelity of the active-site model, and provide confidence in the predictive power of generated docking results.

Atomistic molecular dynamics (MD) simulations were employed to investigate how protein-ligand complexes evolve over time, and if this evolution highlights interactions not predicted by the static docking calculations. MD trajectories were set up and calculated using the Desmond package in Schrödinger's Maestro molecular modeling environment as follows. The input protein-ligand complex was taken from previously generated docking results, and the system was prepared for molecular dynamics using the system builder function and the OPLS3e force field. During this setup the simple point charge (SPC) solvent model was used to solvate an orthorhombic box with buffers of 10 Å between the protein and any boundary of the box. The box volume was minimized, and chloride ions were added to neutralize the system. Chloride ions were excluded from placement within 10 Å of the heme prosthetic group to prevent occupancy of the active site of the enzyme. This resulted in systems of ~50,000 atoms. These prepared systems were then loaded into the molecular dynamics task window and used for the calculation of trajectories with the following parameters. The simulation time was set to 20 ns with a recording interval of 1.2 ps for the energy of the system and 20 ps for the trajectory (atom positions) for a total of 1000 frames recorded. An isothermal-isobaric (NPT) ensemble was used with a temperature of 300 K and a pressure of 1.01325 bar, and the model system was relaxed prior to simulation. Following simulation, analysis was completed using the simulation interactions diagram function within Maestro. Additional parameters are provided in the Supplementary Information.

The area calculations for the size of metal complex protecting groups, which take up 400–515 Å$^3$ of space within the CYP active site, were calculated for Ru(tpy)(6,6'dmbpy) and for Ru(tpy)(bca).

**Reporting summary**. Further information on research design is available in the Nature Research Reporting Summary linked to this article.

## Data availability
The data that support the findings of this study are available within the paper or its Supplementary Information. Source data are provided with this paper, or upon request from the corresponding author. Source data are provided with this paper.

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

## Acknowledgements
We gratefully acknowledge the support of the National Institute of General Medical Sciences of the National Institutes of Health under Award Numbers R01GM138882 and 5R01GM107586.

## Author contributions
E.C.G., D.K.H., and D.H. conceived the project, designed the experiments, and wrote the manuscript. D.H. performed synthesis and photophysical measurements. D.K.H. created the assays, generated all mutants, and performed all biological and biophysical characterization. A.C.H. performed experiments with phLMs, measured singlet oxygen, and performed stability measurements. A.D.F. performed computational calculations. All authors analyzed the data, discussed the results and commented on the manuscript.

## Competing interests
The authors declare no competing interests.
