## [Peer Review File · Nature Communications]

REVIEWER COMMENTS

Reviewer #1 (Remarks to the Author):

The paper describes the creation of a light-activated inhibitor of the cytochrome P450 1B1 (CYP1B1). One of the inhibitors (compound 6) had a selectivity index >100,000 over the closest family member, CYP1A1. Selectivities for CYP1B1 over CYP19A1 and CYPs expressed on human liver microsomes were also high (62,800 and 4,300, respectively). According to the authors, compound 3 is the most selective reported CYP inhibitor, and compound 6 provides the highest photoactivity index described for inhibition of a protein by a Ru(II) complex. The manuscript is well written, but it lacks clarification of some choices made by the authors and it would benefit of a better description of figures and tables.

Major concerns

- Description of computational methods requires improvement.

Page 8, section "Docking and MD experiments.": the parameters for equilibration of the system prior to simulation, and for the simulation itself, are not described (timestep, thermostat, barostat, etc). This makes the work non-reproducible, and makes me wonder if the RMSD values in figure S1 for compound 2 are due to poor equilibration of the system.

Which pH value was used to protonate the protein and the ligands?

Was Glide used with single or extra precision for docking?

- Some choices in the manuscript could be better justified.

a- Page 4, section "Mutational studies": it is not very clear to me how the authors chose the residues to be mutated. Residues 127 and 134 are presented in table S1 as making contacts with one of the inhibitors. Why were residues 269, 332 and 333 selected for mutation?

Why were the other residues in table S1 not used for mutation?

b- Page 4: "Compounds 2 and 3 were combined with Scaffolds I and II to create photocaged CYP1B1 inhibitors, resulting in octahedral Ru(II) polypyridyl coordination complexes 4–6 (Chart 1)."

Results in the manuscript (table 1, figure 1) show that compound 2 has a lower selectivity index compared to compound 3. So it is not clear to me why the authors used compounds 2 and 3 to create photocaged inhibitors, instead of focusing their efforts on compound 3 only.

- Explanation of figures and tables requires improvement.

a- Page 2, figure 1: please revise the label of the x axis in figure 1B. A symbol is missing.

A more complete explanation of the data could be provided, especially for figure 1C.

b- Page 2, table 1: the "I" of IC50 is missing in line 1, column 4.

A more complete explanation of the meaning of every column could be provided.

c- Page 6, figure 3: some symbols are missing in the y axis (figure 3a) and x axis (figure 3d).

A more complete explanation of the data in figure 3e could be provided.

Minor concerns

- Page 3: "Compound 3 was far superior, with an IC50 of 390 pM, making it one of the most potent CYP1B1 inhibitors reported."

The IC50 value shown in table 1 is different. Please revise.

- Page 3: "Regular evaluation over 48 hrs demonstrated the cells were healthy and their growth rates were unaffected."

Are there any data to support this?

How was the healthiness of the cells evaluated?

Please include the growth rates as supporting information.

- Page 3, section "Computational methods"

It would be useful to have a picture of CYP1B1 in the supporting information highlighting the positions of the residues mentioned in table S1.

- Page 4: "However, differences in pKa values for methyl pyrimidine vs. methyl pyridine (2.0 vs. 5.9) may provide a partial explanation for the variation in activity, as the greater basicity of 3 should result in a stronger coordinative bond, and thus, improved potency."

Are these pKa values a prediction or a result known from the literature?

- The IC50 values for compound 2 are not the same in table 1 and table S2. Please revise.

- Page 4: "The prodrug model systems containing pyridine and pyrimidine were stable in media and non-toxic with light exposure, validating the use of the Ru(II) scaffold for photocages."

Are there any data to support this?

- Page 5: "As all CYPs share the same fold, this could result in non-specific inhibition of other CYPs. However, compounds 4 and 5 had no impact on CYP19A1 or CYP1A1 at concentrations $\geq 20 \mu\text{M}$; Ru(bpy) 3 also did not inhibit CYP1B1."

Are there any data to support this?

- Page 5: "As with 4 and 5, the intact complex exhibited inhibition of CYP1B1, but had little effect on other CYPs up to concentrations of $30 \mu\text{M}$ (Figure 2d, Table 1)."

There is no figure 2d.

- Page 6: "..., these metal complexes for us to reassess what we assume about CYP inhibitors."

There is something missing in this sentence.

Reviewer #2 (Remarks to the Author):

This new study by the Glazer team describes the development and evaluation of light-activated inhibitors for cytochrome P450 1B1 based on a their previously reported ruthenium(II) scaffold (Inorg. Chem. 2020, 59, 2, 1006–1013). By incorporating a coordination-site into a known CYP1B1 inhibitor, the authors were able to control enzyme activity with light. Impressively, their design led to a dramatic improvement of the IC50 from 83 nM for the original inhibitor to 0.31 nM when a pyridyl moiety was incorporated. Moreover, when coordinated to the Ru(II)-scaffold, the light-activated "prodrug" exhibits a photoactivity index of $>6,300$ which is extraordinarily high for this type of compounds.

The potency of their new inhibitor, the unprecedented photoindex of the ruthenium-caged version of it, and the relevance of P450s for medicine and biology, fully justifies publication in Nature

Communication. However, before accepting the manuscript, the authors should address the following recommendations:

1) In the introduction, the authors point out the importance of CYPs in relation to various diseases such as cancer. On the other hand, they do not provide biological data related to the effect of their compounds in living cancer cells; ideally, cell viability in a cell line that is resistant to chemotherapy due to drug metabolism by P450 1B1, but would lose its resistance upon light activation of compound 6 and inhibition of P450 1B1. Could the authors comment on the feasibility and relevance of such experiments?

2) The notation ϕ PS used in table 2 and in the text should be more clearly defined. As described in the reported methods, the quantum yield of complex 6 was determined in acetonitrile, while it is compared to the quantum yields of complex 4 and 5 which were measured in water (Table 2, page 5). It is highly probable that performing these experiments in a better coordinating solvent like acetonitrile instead of water will produce different photo-products (the acetonitrile-complex instead of the aqua-complex). Since the photo-substitution kinetics for ruthenium (II) in acetonitrile and water are typically very different, these results should not be directly compared as they are now, but the difference in reaction should be indicated, and the table notes of table 2 should clarify what is shown, and at which wavelength ϕ PC was measured. It is also not clear why the measurement for 6 was made in a different solvent than for 4-5, this could be explained somewhere.

3) The authors report a photo-index $>6,300$ for complex 6 for the inhibition of CYP1B1 when irradiated with red-light (660 nm) which is really impressive. However, there is no data unequivocally demonstrating photorelease of the inhibitor from the photocage using such red light (the photosubstitution data were measured with blue light LEDs at 470 nm, see table 2 and figure S3-S5). Without this data, the red-light induced release of the inhibitor for the complex seems probable (considering Kasha's rule), but it is not proven. The photoreactivity of metal complexes sometimes depends on the irradiation wavelength, for example because of direct population of triplet states, so red light data should be shown and discussed in relation to those obtained with blue light. Table 2 should specify the wavelength used for measuring ϕ PS. Ideally, the blue light values are kept in the paper, for comparison purposes with previous manuscripts, as stated by the authors.

4) On page 5 it is stated that aqua-complex 7 is biologically inert. A reference demonstrating it, or a control cytotoxicity experiment, is needed to support this statement. See also the sentence "the prodrug model systems containing pyridine and pyrimidine were stable in the dark and non-toxic with light exposure, validating the use of the Ru(II) cage scaffold for photocages."

5) The 2 last paragraphs of the part entitled "impact on protein stability" (from "this finding..." to "about CYP inhibitor") read more as part of the discussion than as part of a paragraph on protein stability. I would suggest to move them into the discussion. As the paper actually misses a conclusion (which is included in the discussion) and the sentence "these metal complexes for us to reassess what we assume about CYP inhibitors" reads weird, would it make sense to reorganize the discussion a bit, and to highlight better the final conclusions of the manuscript with a dedicated conclusion paragraph?

6) There are missing synthetic procedures in the supplementary information. Could the authors provide references for the synthesis of Ru(II)-scaffold I or II or describe these? In addition, extra

information (such as peak assignment, integrals) in the provided NMR spectra (Figures S9-S17) would be appreciated.

7) Elemental analysis should be provided for all ruthenium compounds but most importantly for complex 6, as it is the only technique that, in combination with the MS and NMR data provided by the authors, definitely conclude on the analytical purity of new chemicals.

8) Regarding protein stability, have the authors investigated the effect of 7 on the protein stability? It seems that this control is missing.

9) The sentence "This is in an agreement with the photochemical features of Ru(II) complexes with unsubstituted pyridine and diazines as monodentate ligands" will probably be unclear to the non-specialist and could be better explained.

10) Caption of figure 3: the wavelength of the red and blue LED light used should be indicated. Rage should be changed into range. The abbreviation for pooled human liver microsomes used in the insert of Figure 3D (phLM) should be indicated. The wavelengths used in figure 3C for HPLC detection by UV-vis should be indicated. The sentence "the formation of complex 7 and inhibitor 3 were verified by UV/vis" should refer to the proper figure in the ESI.

11) I would suggest rephrasing of the last sentence of the Discussion, as it seems to suggest that the described Ru(II)-complex is a protein inhibitor. For example: "..., and 6 provides the highest reported photoactivity index for a Ru(II)-based photo-caged protein inhibitor."

12) Table 1, 5th column: C50 should be changed into IC50

Reviewer #3 (Remarks to the Author):

The work presented by Havrylyuk et al. describes the discovery of new selective and potent CYP1B1 inhibitors and the process of rendering them photo-responsive.

A known CYP1B1 inhibitor tetramethoxystilbene (TMS) was converted into the coordinating ligand for Ru(II)-based complexes by replacing one dimethoxybenzene moiety with the pyridyl or pyrimidyl group. This structural change yielded inhibitors with properties (selectivity index and potency) superior to any previously known CYP inhibitor. Docking and detailed mutational studies revealed crucial interactions responsible for active-site specific binding.

To achieve a spatiotemporal resolution over the activity of inhibitors, Ru(II)-complexes of inhibitors 2 and 3 were prepared, photochemically and biologically evaluated. While 3 is the most selective and potent CYP inhibitor, the corresponding Ru-complex 6 has shown the highest photoactivity index for Ru(II)-based photocages.

Major comments

- In general, all figure captions in the main text and SI need a more thorough explanation of legends and experimental conditions (reaction times, irradiation conditions, error bars, replicates, etc.)
- Control experiments are scarce. Vehicle controls are omitted in most of the experiments.
- The study did not comment or investigate singlet oxygen formation using Ru(II) complexes as photosensitizers. Triplet sensitization is a common property of these complexes, and it should be a part of the manuscript.
- Is there any benefit of obtaining photo-control over the activity of compound 3 when it already has such a high selectivity index, and CYP1B1 is only overexpressed in the cancer tissue? The manuscript would benefit from in vivo experiments. Keeping in mind the length of setting up in vivo experiments, it is not mandatory for publication, but it is highly recommended. However, since the most important body of this work is built around medicinal chemistry, thus it would be beneficial to test the pharmacokinetics and pharmacodynamics of compounds 2 and 3 and corresponding Ru(II)-complexes. This information would significantly strengthen the paper, and better show the direction for further developing the selective CYP inhibitors.
- Upon bounding to the Ru(II)-complex, coordination of the ligand to the CYP binding site should be prevented. Also, due to the absence of protein-ligand crystal structure, it is necessary to perform more control experiments regarding the background activity of the complex 6. The authors did not elaborate that background activity is potentially a consequence of hydrolysis or dissociation under cellular conditions. It is necessary to test the stability of Ru(II)-complexes:
 - o Under strong reductive conditions in cells – in the presence of a high concentration of GSH
 - o at lower pH values as in specific cellular compartments. The inhibitor could get protonated and consequently dissociated
 - o in the presence of competing coordinating ligands present in cells.
- Cellular viability studies are strongly recommended with all ligands and corresponding complexes in the presence of light and the dark.
- There is a spectrum of the protonated inhibitor 3 in figure 3a? The reasoning is not mentioned at all in the text, and it seems unnecessary.
- Complete UPLC-MS traces are missing in the SI. Only insets are shown.
- Complexes 4–6 exhibit maxima between ~530 and 545 nm in H₂O, with tailing absorption out to 650 nm.

First of all, these are local maxima, and the statement could be misinterpreted for absorption maxima. Secondly, often comments like this are not referred to corresponding figures in the main text and SI.

- The authors mentioned that red light could be used for photo-deprotection, but it is essential to state the light source's specifications. What is full width at half maximum (FWHM), and why a band

or cut-off filter was not used? If 660 nm light was used for the photo-deprotection, removing lower wavelengths with a cut-off filter is crucial.

- Figure S7 and related conclusions: UPLC traces for complexes 4 and 5 are missing, while insets exist only for 6. This is misleading because the authors claim the change in absorbance for complex 6 comes from aggregation while 4 and 5 decompose/hydrolyze. If complexes 4 and 5 hydrolyze, please provide data and consequently change the following sentence in the main text:

However, the potency of the intact complexes (0.2–2 μ M) was striking.

If the complex hydrolyses or decomposes followed by the ligand release, the activity could be attributed to this process, not the complex itself.

General comments

- A stable HEK cell line was created with CYP1B1 expression under the control of tetracycline to facilitate controlled and titratable expression (see the SI for more details)

change SI to Methods section

- Concluding the selectivity based on 38% sequence homology of CYP1B1 and 1A1, and the fact that CYP1B1 and 19A1 are involved in consecutive (and not the same) steps of the metabolic pathway of testosterone, is too strong. It is known that CYP1B1 and 1A1, despite their sequence similarity, have different catalytic efficiencies and produce different metabolites using the same substrates.
- Nowhere in figure 3 is mentioned what light has been used for photo-deprotection.
- The coordination of the 4-substituted pyridyl ligand induced a bathochromic shift of 15 nm, and a tail that extended to 700 nm, facilitating activation with low energy light (Figure 3a).

This sentence comes directly after explaining properties in the water, but it relates to Figure 3a, which shows data in acetonitrile. In water, absorption spectra are blue shifted.

- The stability of each complex was assessed over 24 hours under aqueous conditions at 37 °C. The compounds with the higher Φ PS (4 and 5) exhibited slow degradation over 24 hr, while complex 6 remained stable, with less than 2% degradation (Fig. S7).

Define what does 'aqueous conditions' mean. Also, this is hydrolytic stability in the dark (Fig. S7), and connecting Φ PS and hydrolysis/decomposition/aggregation rate is pointless. Lastly, refer to Fig. S8.

- However, compounds 4 and 5 had no impact on CYP19A1 or CYP1A1 at concentrations 20 μ M; Ru(bpy)₃ also did not inhibit CYP1B1.

Data is missing. Also, it would be better as a control compound to test the Ru(II)-complex used in the paper and not Ru(bpy)₃.

- As with 4 and 5, the intact complex exhibited inhibition of CYP1B1, but had little effect on other CYPs up to concentrations of 30 μ M (Figure 2d, Table 1).

Figure 2d does not exist.

- We are not aware of other inorganic or organic photocages that provide this level of selectivity.

Here, the term 'selectivity' is wrongly used. It can mislead that upon caging, the off-target selectivity has been achieved.

- Addition of carboxylic acids to the biquinoline ligand reduced inhibition of the CYP1B1 enzyme by the complex by ~10-fold (complex 5 vs 4), and concentrations 30 μ M were needed to observe interactions with other CYPs.

Experimental data is missing.

- The thermal stability of CYP1B1 shifted by 3 °C with 10 μ M ANF; a 2 °C stabilization was observed with inhibitor 3 (Table 3, Fig. S20). No shift in stability was seen with 6 at concentrations up to 20 μ M.

The information about the number and type of replicates is missing everywhere. What do the error bars stand for (Table 3, Fig. S20, everywhere else)?

- In the caption of Figure S20 stands Table S1 instead of Table S3
- What remains to be determined is what drives the interaction of metal-coordinated CYP inhibitors with such extraordinary affinity and selectivity to their target enzymes.

This is true, but first, proper control experiments should be conducted.

- Figure S7 should contain information about concentration and the upper solubility limit. All IC50 curves must be revised and adjusted for the maximum soluble concentration if the concentration is too high and leads to precipitation. Provide kinetic solubility test for the applied concentrations.

- Both the CYP and POR expression was under the control of the TetO2 inducible CMV promoter.²

'2' has to be superscript

- CYP1B1 was expressed and purified as reported.

The reference is missing.

The concept of incorporating photoremovable protecting groups is widespread in obtaining light-induced control over the activity of inhibitors, and getting high PI is not uncommon. However, having Ru(II)-based complex of a CYP inhibitor with a high PI imposed the challenge so far. This challenge was solved in this work. On the other hand, a CYP1B1 inhibitor with outstanding potency and selectivity was developed. Compound 3 has some of the pharmacological parameters better than CYP inhibitors already used as drugs. This finding deserves to be published in Nature Communications if the medicinal chemistry part is extended with additional experiments and photochemistry more thoroughly studied.

February 10, 2022

We thank the reviewers for the careful and thoughtful consideration of our work. We have followed all the requests of the reviewers, and now submit our revised manuscript, having strengthened the work by significant edits and additional experiments. Moreover, we agreed with each reviewer's comments and added more detailed and helpful descriptions of the results for each study performed, and explained our rationale for each of the studies so that readers can see the logic behind the various systems developed, and our scientific approach.

Thanks to the reviewers suggestions, our manuscript is now much improved. We also appreciate that the reviewers noted that the compelling results we show and their implications in biology and medicine "fully justifies publication in Nature Communication". We hope the reviewers agree that the report has been greatly enhanced due to the insightful critiques, and can approve its publication.

The full text of the reviewers' comments are below, along with our detailed responses. I am also providing a copy of the manuscript with all the significant changes highlighted.

REVIEWER COMMENTS

Reviewer #1 (Remarks to the Author):

The paper describes the creation of a light-activated inhibitor of the cytochrome P450 1B1 (CYP1B1). One of the inhibitors (compound 6) had a selectivity index $>100,000$ over the closest family member, CYP1A1. Selectivities for CYP1B1 over CYP19A1 and CYPs expressed on human liver microsomes were also high (62,800 and 4,300, respectively). According to the authors, compound 3 is the most selective reported CYP inhibitor, and compound 6 provides the highest photoactivity index described for inhibition of a protein by a Ru(II) complex. The manuscript is well written, but it lacks clarification of some choices made by the authors and it would benefit of a better description of figures and tables.

We thank the reviewer for their comments. We completely agree that clarifications are needed, and we have made significant changes in the text to achieve this. Each point is addressed in detail below.

Major concerns

- Description of computational methods requires improvement.

Page 8, section "Docking and MD experiments.": the parameters for equilibration of the system prior to simulation, and for the simulation itself, are not described (timestep, thermostat, barostat, etc). This makes the work non-reproducible, and makes me wonder if the RMSD values in figure S1 for compound 2 are due to poor equilibration of the system.

All these parameters have now been added to the Supplementary Information (see expanded "Docking and MD experiments" section), and the reader is directed to it in the body of the paper with the sentence "Additional parameters are provided in the Supplemental Information." We used the Schrodinger recommended settings, and don't believe the results are due to poor equilibration.

Which pH value was used to protonate the protein and the ligands? pH 7 +/- 2

Was Glide used with single or extra precision for docking? Extra Precision

- Some choices in the manuscript could be better justified.

a- Page 4, section "Mutational studies": it is not very clear to me how the authors chose the residues to be mutated. Residues 127 and 134 are presented in table S1 as making contacts with one of the inhibitors. Why were residues 269, 332 and 333 selected for mutation?

The reviewer is absolutely right, so we have now added a paragraph explaining our rationale for each mutation.

“The aromatic residues that frame the inhibitor binding site, Phe143, Phe231 and Phe268, were all mutated to Leu, to determine the importance of the π interactions. In addition, Ser127 was chosen for mutation based on interactions identified in the simulations. The Ser127 residue is located in the upper corner of the active site, on the B-C loop. It is anticipated to transiently contact the inhibitors, but also forms a hydrogen bond with Asp326 of the I helix, and thus may impact the position of this helix with regards to the B-C loop. The Ser269 residue, located in the G helix at the top of the active site, was anticipated to form polar contacts with the inhibitor side chains, if the inhibitor disengaged from the heme and slid away against the I helix. Finally, two other active site amino acids, Ala330 and Thr334, are important contacts for inhibitors (Table S1). Rather than directly mutating these residues in the I helix, neighboring residues Gln332 and Asp333 were targeted, with the hope of interrupting amino acid interactions that position the helix within the active site.”

Why were the other residues in table S1 not used for mutation?

Our goal was to provide a illustrative collection of mutants to determine the role of different types of molecular interactions and modulation of protein tertiary contacts. The work required for each mutation led us to prioritize the 6 mutants shown. In most cases, we did not create mutations in neighboring residues. The exceptions to this were Q332 and D333, as we wanted to invert the nature of the neighboring side chains.

b- Page 4: "Compounds 2 and 3 were combined with Scaffolds I and II to create photocaged CYP1B1 inhibitors, resulting in octahedral Ru(II) polypyridyl coordination complexes 4–6 (Chart 1)."

Results in the manuscript (table 1, figure 1) show that compound 2 has a lower selectivity index compared to compound 3. So it is not clear to me why the authors used compounds 2 and 3 to create photocaged inhibitors, instead of focusing their efforts on compound 3 only.

We agree, and have added the following text to clarify why we used both inhibitors (compound 2 is not as appealing as an inhibitor, but we investigated it due to its photochemical superiority).

“It was expected that the diazine ring in compound 2 would provide for superior photochemistry in the Ru(II) complex, as the less basic heterocycle was shown to create complexes with higher quantum yields for photosubstitution, Φ_{PS} . However, the pyridyl ring in 3 was anticipated to produce a more potent CYP inhibitor...”

- Explanation of figures and tables requires improvement.

a- Page 2, figure 1: please revise the label of the x axis in figure 1B. A symbol is missing.

Done.

A more complete explanation of the data could be provided, especially for figure 1C.

Done.

b- Page 2, table 1: the "I" of IC50 is missing in line 1, column 4.

Thank you! Corrected.

A more complete explanation of the meaning of every column could be provided.

Done.

c- Page 6, figure 3: some symbols are missing in the y axis (figure 3a) and x axis (figure 3d).

Thank you! Corrected.

A more complete explanation of the data in figure 3e could be provided.

Thank you! Corrected.

Minor concerns

- Page 3: "Compound 3 was far superior, with an IC₅₀ of 390 pM, making it one of the most potent CYP1B1 inhibitors reported."

The IC₅₀ value shown in table 1 is different. Please revise.

Thank you! Corrected.

- Page 3: "Regular evaluation over 48 hrs demonstrated the cells were healthy and their growth rates were unaffected."

Are there any data to support this?

How was the healthiness of the cells evaluated?

Please include the growth rates as supporting information.

We have added the data in the Supplementary Information, and changed the text in the manuscript: "Regular visual evaluation over 48 hrs indicated the cells were healthy, and their growth rates were unaffected at 72 hrs (at 10 μM), as quantified by metabolically active cells determined by the conversion of resazurin to resorufin (Figures S18–24)."

- Page 3, section "Computational methods"

It would be useful to have a picture of CYP1B1 in the supporting information highlighting the positions of the residues mentioned in table S1.

Thank you for this great suggestion. This is now shown in a three part figure S46.

- Page 4: "However, differences in pK_a values for methyl pyrimidine vs. methyl pyridine (2.0 vs. 5.9) may provide a partial explanation for the variation in activity, as the greater basicity of 3 should result in a stronger coordinative bond, and thus, improved potency."

Are these pK_a values a prediction or a result known from the literature?

They are from a prediction, but the values are consistent with literature reports. This is now addressed in a note (# 36).

- The IC₅₀ values for compound 2 are not the same in table 1 and table S2. Please revise.

Thank you! Corrected.

- Page 4: "The prodrug model systems containing pyridine and pyrimidine were stable in media and non-toxic with light exposure, validating the use of the Ru(II) scaffold for photocages."

Are there any data to support this?

We have modified this statement in the text and added information on the pyridine prodrug model, compound 8. This data is now shown in Figure S24.

- Page 5: "As all CYPs share the same fold, this could result in non-specific inhibition of other CYPs. However, compounds 4 and 5 had no impact on CYP19A1 or CYP1A1 at concentrations \geq 20 μM; Ru(bpy)₃ also did not inhibit CYP1B1."

Are there any data to support this?

We removed reference to Ru(bpy)₃ and replaced it with compound **8**, which is a better model for **5** and **6**. The data for **4** and **5** are in Figures S21 and S22. The text has been changed to the following: "The control compound, **8**, which has the same scaffold as **5** and **6**, but releases pyridine, also had no impact on any CYP at concentrations up to 30 μM both in the dark and following irradiation to form **7**."

- Page 5: "As with **4** and **5**, the intact complex exhibited inhibition of CYP1B1, but had little effect on other CYPs up to concentrations of 30 uM (Figure 2d, Table 1)."

There is no figure 2d.

Thank you! Corrected.

- Page 6: "..., these metal complexes for us to reassess what we assume about CYP inhibitors."

There is something missing in this sentence.

Thank you! Corrected.

"...these metal complexes force us to reassess what we assume about CYP inhibitors."

Reviewer #2 (Remarks to the Author):

This new study by the Glazer team describes the development and evaluation of light-activated inhibitors for cytochrome P450 1B1 based on a their previously reported ruthenium(II) scaffold (Inorg. Chem. 2020, 59, 2, 1006–1013). By incorporating a coordination-site into a known CYP1B1 inhibitor, the authors were able to control enzyme activity with light. Impressively, their design led to a dramatic improvement of the IC₅₀ from 83 nM for the original inhibitor to 0.31 nM when a pyridyl moiety was incorporated. Moreover, when coordinated to the Ru(II)-scaffold, the light-activated "prodrug" exhibits a photoactivity index of >6,300 which is extraordinarily high for this type of compounds.

The potency of their new inhibitor, the unprecedented photoindex of the ruthenium-caged version of it, and the relevance of P450s for medicine and biology, fully justifies publication in Nature Communication.

We thank the reviewer!

However, before accepting the manuscript, the authors should address the following recommendations:

1) In the introduction, the authors point out the importance of CYPs in relation to various diseases such as cancer. On the other hand, they do not provide biological data related to the effect of their compounds in living cancer cells; ideally, cell viability in a cell line that is resistant to chemotherapy due to drug metabolism by P450 1B1, but would lose its resistance upon light activation of compound **6** and inhibition of P450 1B1. Could the authors comment on the feasibility and relevance of such experiments?

There are several pre-clinical reports demonstrating that CYP1B1 inhibitors suppress chemotherapeutic resistance in cancer cells. However, the cell biology of CYP1B1 is known to be complex, and unfortunately, we do not have an appropriate model system in place to test this yet. The model system should be a cancer cell line with inducible and titratable expression of CYP1B1, that can be grown in 3D tumor models and implanted *in vivo*, as several of the effects of CYP1B1 appear related to invasion and metastasis. We are working diligently with a collaborator who is a cancer biologist to achieve this for future studies.

2) The notation phiPS used in table 2 and in the text should be more clearly defined.

Done.

As described in the reported methods, the quantum yield of complex 6 was determined in acetonitrile, while it is compared to the quantum yields of complex 4 and 5 which were measured in water (Table 2, page 5). It is highly probable that performing these experiments in a better coordinating solvent like acetonitrile instead of water will produce different photo-products (the acetonitrile-complex instead of the aqua-complex). Since the photo-substitution kinetics for ruthenium (II) in acetonitrile and water are typically very different, these results should not be directly compared as they are now, but the difference in reaction should be indicated, and the table notes of table 2 should clarify what is shown, and at which wavelength ϕ_{PC} was measured. It is also not clear why the measurement for 6 was made in a different solvent than for 4-5, this could be explained somewhere.

We completely agree, and apologize for the oversight and lack of explanation. This section has been significantly expanded and more photochemical information is provided in a number of solutions.

We now provide ϕ_{PS} in water with 5% DMSO for all compounds so they may be compared to each other; DMSO had to be used with compound 6 as it suffered from solubility issues in the absence of DMSO. Unfortunately, the HPLC method could not be used for Opti-MEM solutions, so this data is not available for compound 6. However, we added values in methanol, as this is a poorer coordinating solvent than acetonitrile. The reviewer is correct, the values are very environmentally sensitive, and we have added a comment to this effect:

“The Φ_{PS} for 6 is also environmentally sensitive, with higher values in less polar environments (Table S7), which may be due to improved solubility.”

3) The authors report a photo-index $>6,300$ for complex 6 for the inhibition of CYP1B1 when irradiated with red-light (660 nm) which is really impressive. However, there is no data unequivocally demonstrating photorelease of the inhibitor from the photocage using such red light (the photosubstitution data were measured with blue light LEDs at 470 nm, see table 2 and figure S3-S5). Without this data, the red-light induced release of the inhibitor for the complex seems probable (considering Kasha's rule), but it is not proven. The photoreactivity of metal complexes sometimes depends on the irradiation wavelength, for example because of direct population of triplet states, so red light data should be shown and discussed in relation to those obtained with blue light.

We agree with the reviewer, and apologize that we were not clear that all the photochemical data in Figure 3 is with irradiation with red light! This is now more clearly defined.

Table 2 should specify the wavelength used for measuring ϕ_{PS} . Ideally, the blue light values are kept in the paper, for comparison purposes with previous manuscripts, as stated by the authors.

Done.

4) On page 5 it is stated that aqua-complex 7 is biologically inert. A reference demonstrating it, or a control cytotoxicity experiment, is needed to support this statement. See also the sentence “the prodrug model systems containing pyridine and pyrimidine were stable in the dark and non-toxic with light exposure, validating the use of the Ru(II) cage scaffold for photocages.”

We apologize for the oversight. The data for cytotoxicity for this compound and for enzyme inhibition are now shown in Figure S24, and referenced in the body of the text. Please see above.

5) The 2 last paragraphs of the part entitled “impact on protein stability” (from “this finding...” to “about CYP inhibitor”) read more as part of the discussion than as part of a paragraph on protein stability. I would suggest to move them into the discussion. As the paper actually misses a

conclusion (which is included in the discussion) and the sentence “these metal complexes for us to reassess what we assume about CYP inhibitors” reads weird, would it make sense to reorganize the discussion a bit, and to highlight better the final conclusions of the manuscript with a dedicated conclusion paragraph?

We thank the reviewer for this helpful suggestion, and have implemented the recommended change. We have also reworked our conclusion section.

6) There are missing synthetic procedures in the supplementary information. Could the authors provide references for the synthesis of Ru(II)-scaffold I or II or describe these? In addition, extra information (such as peak assignment, integrals) in the provided NMR spectra (Figures S9-S17) would be appreciated.

Corrected, thank you.

7) Elemental analysis should be provided for all ruthenium compounds but most importantly for complex 6, as it is the only technique that, in combination with the MS and NMR data provided by the authors, definitely conclude on the analytical purity of new chemicals.

We respectfully disagree with this comment, as we regularly observe solvent in our crystal structures, and the solvent alters our elemental analysis results. Due to this experience, we feel that the HPLC data is more meaningful, and have provided the data with detection at three additional wavelengths that were chosen to ensure the most sensitive detection of the most likely impurities (see Figure S44, where we show the raw data). However, we also submitted the samples to a contract company for elemental analysis, and now have included this data.

8) Regarding protein stability, have the authors investigated the effect of 7 on the protein stability? It seems that this control is missing.

We ran this control and added the data to Table 3. There was no change in the melting temperature – an interesting contrast to the small decrease observed with complex 6.

9) The sentence “This is in an agreement with the photochemical features of Ru(II) complexes with unsubstituted pyridine and diazines as monodentate ligands” will probably be unclear to the non-specialist and could be better explained.

We thank the reviewer and agree, and we have clarified the sentence.

10) Caption of figure 3: the wavelength of the red and blue LED light used should be indicated.

Done. Range should be changed into range. Thank you! The abbreviation for pooled human liver microsomes used in the insert of Figure 3D (phLM) should be indicated. Done. The wavelengths used in figure 3C for HPLC detection by UV-vis should be indicated. Done. The sentence “the formation of complex 7 and inhibitor 3 were verified by UV/vis” should refer to the proper figure in the ESI. Done.

11) I would suggest rephrasing of the last sentence of the Discussion, as it seems to suggest that the described Ru(II)-complex is a protein inhibitor. For example: “..., and 6 provides the highest reported photoactivity index for a Ru(II)-based photo-caged protein inhibitor.”

Thank you, we have implemented this suggestion.

12) Table 1, 5th column: C50 should be changed into IC50

Thank you, done.

Reviewer #3 (Remarks to the Author):

The work presented by Havrylyuk et al. describes the discovery of new selective and potent CYP1B1 inhibitors and the process of rendering them photo-responsive.

A known CYP1B1 inhibitor tetramethoxystilbene (TMS) was converted into the coordinating ligand for Ru(II)-based complexes by replacing one dimethoxybenzene moiety with the pyridyl or pyrimidyl group. This structural change yielded inhibitors with properties (selectivity index and potency) superior to any previously known CYP inhibitor. Docking and detailed mutational studies revealed crucial interactions responsible for active-site specific binding.

To achieve a spatiotemporal resolution over the activity of inhibitors, Ru(II)-complexes of inhibitors 2 and 3 were prepared, photochemically and biologically evaluated. While 3 is the most selective and potent CYP inhibitor, the corresponding Ru-complex 6 has shown the highest photoactivity index for Ru(II)-based photocages.

Major comments

- In general, all figure captions in the main text and SI need a more thorough explanation of legends and experimental conditions (reaction times, irradiation conditions, error bars, replicates, etc.)

Done.

- Control experiments are scarce. Vehicle controls are omitted in most of the experiments.

We apologize for our lack of clarity. Vehicle controls were run in each experiment, and provide the information to allow for normalization of the data. We have added the information for this process to the Methods section to detail the fact that a “no compound” control is used for 100% activity (or viability, in the cell cytotoxicity experiments) and the “no cell” control is used to provide the “zero” value. Positive control experiments are performed with known inhibitors, and compound **8** has been added as a negative control.

- The study did not comment or investigate singlet oxygen formation using Ru(II) complexes as photosensitizers. Triplet sensitization is a common property of these complexes, and it should be a part of the manuscript.

The reviewer is right. We have tested the ability of the compounds to produce singlet oxygen with 660 nm light, and they do not create a significant amount. See Figure S27.

- Is there any benefit of obtaining photo-control over the activity of compound 3 when it already has such a high selectivity index, and CYP1B1 is only overexpressed in the cancer tissue? The manuscript would benefit from *in vivo* experiments. Keeping in mind the length of setting up *in vivo* experiments, it is not mandatory for publication, but it is highly recommended. However, since the most important body of this work is built around medicinal chemistry, thus it would be beneficial to test the pharmacokinetics and pharmacodynamics of compounds 2 and 3 and corresponding Ru(II)-complexes. This information would significantly strengthen the paper, and better show the direction for further developing the selective CYP inhibitors.

Please see the comments related to a similar question from Reviewer 2. While we ultimately intend to achieve demonstration of *in vivo* efficacy, this is outside our area of expertise and we are working with a collaborator who is a cancer biologist and can assist us with *in vivo* experiments. We thank the reviewer for noting that the *in vivo* experiments are not required for publication. Meanwhile, we have recently obtained pharmacokinetic information for the compounds, which is now shown in Tables S4-6. It shows that one potential benefit of creating the Ru-protected system is that metal coordination slows the degradation of the inhibitors. (Please see results for the Os(II) complex provided; we used a contract company that is unfamiliar with systems with photoprotecting groups, so we could not have them test the light-sensitive Ru(II) complexes).

- Upon bounding to the Ru(II)-complex, coordination of the ligand to the CYP binding site should be prevented. Also, due to the absence of protein-ligand crystal structure, it is necessary to perform more control experiments regarding the background activity of the complex 6. The authors did not elaborate that background activity is potentially a consequence of hydrolysis or dissociation under cellular conditions. It is necessary to test the stability of Ru(II)-complexes:

- o Under strong reductive conditions in cells – in the presence of a high concentration of GSH

- o at lower pH values as in specific cellular compartments. The inhibitor could get protonated and consequently dissociated

- o in the presence of competing coordinating ligands present in cells.

This was a concern we shared. We have added the data presented in Figure S16-17, showing stability in the presence of GSH, imidazole, and at pH = 1. We have added text to the manuscript explaining the possibility of hydrolysis. We also believe that if liberation of the inhibitor were occurring, we would expect the IC₅₀ value in the dark to decrease with increasing incubation time (which is not observed).

We were further convinced that the intact complexes are enzyme inhibitors by data from compounds not included in this manuscript. We synthesized osmium analogues of light-activated Ru(II) agents, as the Os(II) systems are stable and the ligands do not dissociate over a period of weeks. These intact complexes are also potent inhibitors (IC₅₀ values from 0.2-2 μM).

- Cellular viability studies are strongly recommended with all ligands and corresponding complexes in the presence of light and the dark.

This has been done; please see response to Reviewers above.

- There is a spectrum of the protonated inhibitor 3 in figure 3a? The reasoning is not mentioned at all in the text, and it seems unnecessary.

We apologize; we now clarify that this is provided as the UV/Vis obtained from the HPLC showed the protonated form due to the acid in the mobile phase.

- Complete UPLC-MS traces are missing in the SI. Only insets are shown.

The mass spec analysis was performed by direct injection, so there are no chromatograms to show. We are thus providing the HPLC chromatograms for all compounds, and Figure S44 shows the HPLC chromatograms with three additional detection wavelengths.

- Complexes 4–6 exhibit maxima between ~530 and 545 nm in H₂O, with tailing absorption out to 650 nm.

First of all, these are local maxima, and the statement could be misinterpreted for absorption maxima. Secondly, often comments like this are not referred to corresponding figures in the main text and SI.

The reviewer is correct, so we have now specified that this is the local maxima for the MLCT, and ensured that the reader is directed to figures and tables throughout the manuscript.

- The authors mentioned that red light could be used for photo-deprotection, but it is essential to state the light source's specifications. What is full width at half maximum (FWHM), and why a band or cut-off filter was not used? If 660 nm light was used for the photo-deprotection, removing lower wavelengths with a cut-off filter is crucial.

We have added the emission spectrum of the LED (Figure S45) and a statement in the Methods section about the FWHM value, which is only 20 nm. As we used LED light sources, which provide a narrow range of wavelengths, cut off filters are not needed.

- Figure S7 and related conclusions: UPLC traces for complexes 4 and 5 are missing, while insets exist only for 6. This is misleading because the authors claim the change in absorbance for complex 6 comes from aggregation while 4 and 5 decompose/hydrolyze. If complexes 4 and 5 hydrolyze, please provide data and consequently change the following sentence in the main text: However, the potency of the intact complexes (0.2–2 μ M) was striking. If the complex hydrolyses or decomposes followed by the ligand release, the activity could be attributed to this process, not the complex itself.

Please see Figure S13-14 for HPLC traces for 4 and 5.

General comments

- A stable HEK cell line was created with CYP1B1 expression under the control of tetracycline to facilitate controlled and titratable expression (see the SI for more details)
change SI to Methods section

- Concluding the selectivity based on 38% sequence homology of CYP1B1 and 1A1, and the fact that CYP1B1 and 19A1 are involved in consecutive (and not the same) steps of the metabolic pathway of testosterone, is too strong. It is known that CYP1B1 and 1A1, despite their sequence similarity, have different catalytic efficiencies and produce different metabolites using the same substrates.

We agree with the reviewer that these different CYPs do perform different chemistries with different efficiencies, but we see them as the most logical points of comparison to CYP1B1, given current knowledge in the field. The nearest family member is commonly evaluated, which is why we chose CYP1A1, and investigating a related system that share substrates or products in common also seemed like an opportunity for assessing potentially different binding modes. We also chose CYP1A1 and 19A1 as they are implicated in cancers, so we felt that the data provided with the different inhibitors might offer insights for these important CYPs. We hope this answers the reviewer's comment, but we are uncertain that we have understood what they are requesting. We are open to additional improvements with more direction on this point.

- Nowhere in figure 3 is mentioned what light has been used for photo-deprotection.

We apologize for the oversight! The 660 nm light was used for all the photoactivated data in Figure 3; this is now stated (“...following irradiation with 660 nm light (58.7 J/cm²).”)

- The coordination of the 4-substituted pyridyl ligand induced a bathochromic shift of 15 nm, and a tail that extended to 700 nm, facilitating activation with low energy light (Figure 3a).

This sentence comes directly after explaining properties in the water, but it relates to Figure 3a, which shows data in acetonitrile. In water, absorption spectra are blue shifted.

The reviewer is right, so we have replaced the UV/Vis spectra in Figure 3a with data taken in water.

- The stability of each complex was assessed over 24 hours under aqueous conditions at 37 °C. The compounds with the higher Φ PS (4 and 5) exhibited slow degradation over 24 hr, while complex 6 remained stable, with less than 2% degradation (Fig. S7).

Define what does 'aqueous conditions' mean. Also, this is hydrolytic stability in the dark (Fig. S7), and connecting Φ PS and hydrolysis/decomposition/aggregation rate is pointless. Lastly, refer to Fig. S8.

We conducted the experiments in both water and Opti-MEM with 2% FBS to better mimic the cellular conditions. This is now clearly explained in Figure S12, and the reader is directed to this data.

We agree with the reviewer; in principle that the hydrolytic stability in the dark is not correlated to the Φ_{PS} , but it has been our experience that they are related. We showed an inverse correlation between these values in Reference 38 (Havrylyuk, D., Stevens, K., Parkin, S. & Glazer, E. C. Toward Optimal Ru(II) Photocages: Balancing Photochemistry, Stability, and Biocompatibility Through Fine Tuning of Steric, Electronic, and Physicochemical Features. *Inorg. Chem.* **59**, 1006-1013, doi:10.1021/acs.inorgchem.9b02065 (2020).) In the current text, we are only reporting the results we observe, and are not providing any theory or rationale (we don't have one yet).

- However, compounds 4 and 5 had no impact on CYP19A1 or CYP1A1 at concentrations 20 μ M; Ru(bpy)₃ also did not inhibit CYP1B1.

Data is missing. Also, it would be better as a control compound to test the Ru(II)-complex used in the paper and not Ru(bpy)₃.

We agree, and this has now been corrected. Please see a more detailed discussion of this point above.

- As with 4 and 5, the intact complex exhibited inhibition of CYP1B1, but had little effect on other CYPs up to concentrations of 30 μ M (Figure 2d, Table 1).

Figure 2d does not exist.

This is now corrected, thank you!

- We are not aware of other inorganic or organic photocages that provide this level of selectivity. Here, the term 'selectivity' is wrongly used. It can mislead that upon caging, the off-target selectivity has been achieved.

We agree. We have changed “selectivity” to “photocontrol”, as we were discussing the PI value of 6,300.

- Addition of carboxylic acids to the biquinoline ligand reduced inhibition of the CYP1B1 enzyme by the complex by ~10-fold (complex 5 vs 4), and concentrations 30 μ M were needed to observe interactions with other CYPs.

Experimental data is missing.

This has been corrected, see Figure S22.

- The thermal stability of CYP1B1 shifted by 3 °C with 10 μ M ANF; a 2 °C stabilization was observed with inhibitor 3 (Table 3, Fig. S20). No shift in stability was seen with 6 at concentrations up to 20 μ M.

The information about the number and type of replicates is missing everywhere. What do the error bars stand for (Table 3, Fig. S20, everywhere else)?

To ensure rigor and reproducibility, we retested all compounds, and added control compound **8** with light exposure (thus producing compound **7**). We replaced the values in the text with the new values, and the error bars reflect the standard deviation from at least two independent experiments. The quality of the data is excellent, increasing confidence in the findings. While the shifts are small, the data is highly reproducible, as reflected by duplicate experiments, which were performed over a period of months.

- In the caption of Figure S20 stands Table S1 instead of Table S3

This is now corrected, thank you!

- What remains to be determined is what drives the interaction of metal-coordinated CYP inhibitors

with such extraordinary affinity and selectivity to their target enzymes.

This is true, but first, proper control experiments should be conducted.

We hope the reviewer feels the sentence is satisfactory given the control compounds and control systems studies.

- Figure S7 should contain information about concentration and the upper solubility limit. All IC₅₀ curves must be revised and adjusted for the maximum soluble concentration if the concentration is too high and leads to precipitation. Provide kinetic solubility test for the applied concentrations.

We have done this, and the kinetic solubility data is now shown in Table S3. In all cases, we kept the tested concentrations for the IC₅₀ curves below the solubility limits.

- Both the CYP and POR expression was under the control of the TetO² inducible CMV promoter. '2' has to be superscript

This is now corrected, thank you!

- CYP1B1 was expressed and purified as reported.

The reference is missing.

This is now corrected, thank you!

The concept of incorporating photoremovable protecting groups is widespread in obtaining light-induced control over the activity of inhibitors, and getting high PI is not uncommon. However, having Ru(II)-based complex of a CYP inhibitor with a high PI imposed the challenge so far. This challenge was solved in this work. On the other hand, a CYP1B1 inhibitor with outstanding potency and selectivity was developed. Compound 3 has some of the pharmacological parameters better than CYP inhibitors already used as drugs. This finding deserves to be published in Nature Communications if the medicinal chemistry part is extended with additional experiments and photochemistry more thoroughly studied.

Thank you, we really appreciate the comments.

REVIEWER COMMENTS

Reviewer #1 (Remarks to the Author):

I thank the authors of the manuscript for addressing my concerns. The readability of the manuscript is improved, and the computational methods are better described, allowing reproducibility of the work. I still have minor concerns before publication.

Minor concerns

- Page 3: "Regular visual evaluation over 48 hrs indicated the cells were healthy, and their viability and growth rates were unaffected at 72 hrs with 10 μ M compound, as quantified by metabolically active cells determined by the conversion of resazurin to resorufin (Figures S18–20)."

This is an overstatement: cell viability was reduced with 10 μ M compound in figures S18 and S19. No data for growth rates is shown. Please rephrase.

- The description of figures S18-S20 could be improved, with more details about the experiments.

- Since the work aims at application of the compounds in medicinal chemistry, it would be useful to know if the designed compounds follow Lipinski's rule of five.

Reviewer #2 (Remarks to the Author):

All my comments have been addressed convincingly by the authors, who have done a very serious revision. From the HPLC chromatograms I see the compounds are pure and indeed react with 660 nm light, which is quite impressive, as the rest of the paper by the way. The discussion and conclusion have been re-written and are now very convincing as well, the article should definitively be accepted by Nature Communications after the last (minor) comments below have been addressed.

First, I do have an additional comment about C,H,N analysis: it is completely normal to observe solvent in crystal structures, and to include such lattice solvents in the fitting of an elemental analysis. This should by no means prevent considering C,H,N analysis as the only analytical method capable to prove the chemical purity of a sample, and in particular the absence of a salt, which HPLC cannot address. In principle, if the composition $[6]2PF_6 \cdot 2CH_3CN \cdot 5C_3H_6O$ fits with the observed C,H,N composition, it is perfectly fine to report it like the authors did. However, I do wonder where the MeCN comes from, as according to the experimental procedure no MeCN has been used during the synthesis of this compound. Did the authors forget to report some washing or recrystallization or HPLC purification procedure that used acetonitrile? If not, then the fitting might need to be adjusted, with a slightly less good fit maybe, but only using solvents that have been effectively used during synthesis. Same comment for the analysis of compound 8. I do notice that the fitting of compound 3 and 4 fit perfectly well.

Another remark about the statement that “the photosubstitution QY for 6 is environmentally sensitive, with higher values in less polar environments, may be due to improved solubility”. Could another argument explain this observation? The 3MLCT states involved in the photochemistry of such compounds are notoriously sensitive to the polarity of the environment: they are stabilized in polar environments due to their charge transfer character, hence de-stabilized in apolar environments. The 3MC states do not show such dependence, because they are not charge-transfer states. Hence, the 3MLCT-3MC gap should decrease in apolar environment due to the relative destabilization of the 3MLCT states, compared to the 3MC states, which would lead to improved photosubstitution QY. This argument would be more convincing to me than the argument of solubility.

Finally, a few minor remarks and corrections that should be considered before publication:

- Color coding of figure 3a is missing: which curve is which condition?
- the IUPAC does not recommend anymore the use of the Einstein unit for naming a mol of photon. Mol should be used instead. The units of the photon flux measurements should be adapted accordingly (eg figure s7).
- Figure s11: it would help the reader to define compound 7 for the red trace, as it is only shown in figure s24 later in the ESI.
- Figure s12: why showing compounds 5,4,6 in that order for a,b,c, and not 4,5,6?
- Figure s13-s14: any idea what the decomposition products are?
- Figure s18-s20 and figure 3: “% activity” in the y-axis is a bit unclear, I would suggest “% enzyme activity”, as for cell growing inhibition, which these curves are not showing, one often finds %activity as well one the y axis (here the authors use “%viability”). Idem in the caption: “dose response” throughout the manuscript is unclear, as it is also an expression used for cell growth inhibition. When both enzyme inhibition and cell growth inhibition are shown in the same manuscript, I would

suggest to systematically make a clear distinction between both types of dose-response curves. Same idea for the terms “potency” and “photoindex (PI) value” in the caption of figure 3: which potency and which PI value, that for enzyme activity inhibition, or that for cell killing/growth inhibition? As a side note, the definition of the PI value in the text is incorrect, I read at the bottom of the page containing figure 3: “(PI; the ratio of the IC50 values in the light and dark)”, while PI is usually defined as $IC_{50,dark}/IC_{50,light}$.

- Figure s26: can the authors explain “The data for 8 falls under the “no compound” data”?

- Figure s27: the Ru(bpy)₃ complex is not really a proper positive control as it does not absorb red light, so the intensity of the 1O₂ generation cannot be compared to that in the other conditions. Usually, 1O₂ generation quantum yields are calculated. Could the authors find a positive control that is compatible with their irradiation conditions?

- Figure S39-s40: the calculated m/z should include the protonation state, thus if the ligand C₁₄H₁₄N₂O₂ has a calculated mass of 242.1055, then the species observed is protonated and the calculated m/z should not be 242, 10, but 243.11. Check in all captions of all mass spectra and experimental part.

- Table S7: I am not sure from which figures these data derive, ie if they derive from figures s3-s5 (UV-vis analysis), or from figures s6-s10 (HPLC analysis). This information should be indicated. If they are from UV-vis data, I wonder how it is possible to measure the values in Opti-MEM: usually in such medium several photoproducts are possible, considering the large number of ligands that can potentially bind to ruthenium after photochemical cleavage of the Ru-N bond. In such a case we might see an “apparent” QY for a range of photochemical reactions taking place simultaneously. By contrast, HPLC QY are in principle possible to measure if one follows the peak of the reagent vs. the amount of mole of photon absorbed since t=0.

Reviewer #3 (Remarks to the Author):

I am very pleased by the thorough and thoughtful revision of the manuscript and thus recommend publication.

Response to Review

We appreciate the efforts of the reviewers, and have made each of their requested changes. Please find the reviewers' comments below, with responses in red.

Reviewer #1 (Remarks to the Author):

I thank the authors of the manuscript for addressing my concerns. The readability of the manuscript is improved, and the computational methods are better described, allowing reproducibility of the work. (Thank you!) I still have minor concerns before publication.

Minor concerns

- Page 3: "Regular visual evaluation over 48 hrs indicated the cells were healthy, and their viability and growth rates were unaffected at 72 hrs with 10 μ M compound, as quantified by metabolically active cells determined by the conversion of resazurin to resorufin (Figures S18–20)."

This is an overstatement: cell viability was reduced with 10 μ M compound in figures S18 and S19. No data for growth rates is shown. Please rephrase.

We have altered this text so that we provide a quantitative value for viability and removed the comment about growth rates. We added details on the normalization process and the cellular doubling rate to the figure captions for Figures S18-20 so that the reader will be able to consider the impact on cell division.

- The description of figures S18-S20 could be improved, with more details about the experiments. This was completed.

- Since the work aims at application of the compounds in medicinal chemistry, it would be useful to know if the designed compounds follow Lipinski's rule of five. We now provide this analysis in Table S8, which was added to SI, and direct the reader to it in a note (56).

Reviewer #2 (Remarks to the Author):

All my comments have been addressed convincingly by the authors, who have done a very serious revision. From the HPLC chromatograms I see the compounds are pure and indeed react with 660 nm light, which is quite impressive, as the rest of the paper by the way. The discussion and conclusion have been re-written and are now very convincing as well, the article should definitively be accepted by Nature Communications after the last (minor) comments below have been addressed.

Thank you, we really appreciate your comments.

First, I do have an additional comment about C,H,N analysis: it is completely normal to observe solvent in crystal structures, and to include such lattice solvents in the fitting of an elemental analysis. This should by no means prevent considering C,H,N analysis as the only analytical method capable to prove the chemical purity of a sample, and in particular the absence of a salt, which HPLC cannot address. In principle, if the composition $[6]2PF_6 \cdot 2CH_3CN \cdot 5C_3H_6O$ fits with the observed C,H,N composition, it is perfectly fine to report it like the authors did. However, I do wonder where the MeCN comes from, as according to the experimental procedure no MeCN has been used during the synthesis of this compound. Did the authors forget to report some washing or recrystallization or HPLC purification procedure that used acetonitrile? If not, then the fitting might need to be adjusted, with a slightly less

good fit maybe, but only using solvents that have been effectively used during synthesis. Same comment for the analysis of compound 8. I do notice that the fitting of compound 3 and 4 fit perfectly well.

This was corrected and edited. We thank the reviewer for noticing this, and revised the experimental description, which had stated “acetone” when it should have stated “acetonitrile”.

Another remark about the statement that “the photosubstitution QY for 6 is environmentally sensitive, with higher values in less polar environments, may be due to improved solubility”. Could another argument explain this observation? The 3MLCT states involved in the photochemistry of such compounds are notoriously sensitive to the polarity of the environment: they are stabilized in polar environments due to their charge transfer character, hence de-stabilized in apolar environments. The 3MC states do not show such dependence, because they are not charge-transfer states. Hence, the 3MLCT-3MC gap should decrease in apolar environment due to the relative destabilization of the 3MLCT states, compared to the 3MC states, which would lead to improved photosubstitution QY. This argument would be more convincing to me than the argument of solubility. We agree that this is a feasible explanation for our result, so we have added this to note 45. Unfortunately, detection of the energy of the dark 3MLCT state is not possible for us, as it would require advanced spectroscopy techniques such as transient absorption.

Finally, a few minor remarks and corrections that should be considered before publication:

- Color coding of figure 3a is missing: which curve is which condition? This was revised and edited.
- the IUPAC does not recommend anymore the use of the Einstein unit for naming a mol of photon. Mol should be used instead. The units of the photon flux measurements should be adapted accordingly (eg figure s7). This was revised and edited.
- Figure s11: it would help the reader to define compound 7 for the red trace, as it is only shown in figure s24 later in the ESI. This was revised and edited.
- Figure s12: why showing compounds 5,4,6 in that order for a,b,c, and not 4,5,6? This was revised and edited.
- Figure s13-s14: any idea what the decomposition products are? This was revised and edited.
- Figure s18-s20 and figure 3: “% activity” in the y-axis is a bit unclear, I would suggest “% enzyme activity”, as for cell growing inhibition, which these curves are not showing, one often finds %activity as well on the y axis (here the authors use “%viability”). Idem in the caption: “dose response” throughout the manuscript is unclear, as it is also an expression used for cell growth inhibition. When both enzyme inhibition and cell growth inhibition are shown in the same manuscript, I would suggest to systematically make a clear distinction between both types of dose-response curves. Same idea for the terms “potency” and “photoindex (PI) value” in the caption of figure 3: which potency and which PI value, that for enzyme activity inhibition, or that for cell killing/growth inhibition? As a side note, the definition of the PI value in the text is incorrect, I read at the bottom of the page containing figure 3: “(PI; the ratio of the IC50 values in the light and dark)”, while PI is usually defined as $IC_{50,dark}/IC_{50,light}$. This was revised and edited.
- Figure s26: can the authors explain “The data for 8 falls under the “no compound” data”? This was revised and edited by using a dashed line.

- Figure s27: the Ru(bpy)₃ complex is not really a proper positive control as it does not absorb red light, so the intensity of the 1O₂ generation cannot be compared to that in the other conditions. Usually, 1O₂ generation quantum yields are calculated. Could the authors find a positive control that is compatible with their irradiation conditions? **This was achieved by using Photochlor as a control.**

- Figure S39-s40: the calculated m/z should include the protonation state, thus if the ligand C₁₄H₁₄N₂O₂ has a calculated mass of 242.1055, then the species observed is protonated and the calculated m/z should not be 242, 10, but 243.11. Check in all captions of all mass spectra and experimental part. **This was revised and edited.**

- Table S7: I am not sure from which figures these data derive, ie if they derive from figures s3-s5 (UV-vis analysis), or from figures s6-s10 (HPLC analysis). This information should be indicated. If they are from UV-vis data, I wonder how it is possible to measure the values in Opti-MEM: usually in such medium several photoproducts are possible, considering the large number of ligands that can potentially bind to ruthenium after photochemical cleavage of the Ru-N bond. In such a case we might see an “apparent” QY for a range of photochemical reactions taking place simultaneously. By contrast, HPLC QY are in principle possible to measure if one follows the peak of the reagent vs. the amount of mole of photon absorbed since t=0. **This was revised and edited in SI. The reviewer is correct, so we added a note to indicate this to Table S7.**

Reviewer #3 (Remarks to the Author):

I am very pleased by the thorough and thoughtful revision of the manuscript and thus recommend publication.

Thank you!!!